# ON CODE-INDUCED REASONING IN LLMS

**Abdul Waheed**[*]  **Zhen Wu**[*]  **Carolyn Rosé**   **Daphne Ippolito**
Carnegie Mellon University
{abdulw,zhenwu,cprose,dippolit}@cs.cmu.edu

 Code    Data

## ABSTRACT

Code data has been shown to enhance the reasoning capabilities of large language models (LLMs), but it remains unclear which aspects of code are most responsible. We investigate this question with a systematic, data-centric framework. We construct parallel instruction datasets across ten programming languages and introduce controlled perturbations that selectively disrupt structural and semantic properties of code. We then fine-tune LLMs from five model families and eight scales on each variant and evaluate their performance on natural language, math, and code tasks. Across 3,331 experiments, our results show that LLMs are more vulnerable to structural perturbations than semantic ones, particularly on math and code tasks. Appropriate abstractions like pseudocode and flowcharts can be as effective as code, while encoding the same information with fewer tokens without adhering to original syntax can often retain or even improve performance. Notably, even corrupted code with misleading signals remains competitive when surface-level regularities persist. Finally, syntactic styles also shape task-specific gains, with Python favoring natural language reasoning and lower-level languages such as Java and Rust favoring math. Through our systematic framework, we provide a fine-grained analysis of how different aspects of code influence reasoning and inform the design of training data for enhancing LLM reasoning capabilities.

## 1 INTRODUCTION

There has been substantial interest in the last several years in engineering language models that can tackle challenging reasoning tasks (Huang & Chang, 2023; Xu et al., 2025; Wei et al., 2022; Kojima et al., 2023). Language reasoning tasks, such as math word problems or logic puzzles, tend to require multi-step, structured "thinking" in order to produce the correct answer. Recent work has found that training the language model on code, either during pre-training (Fu & Khot, 2022; Ma et al., 2023b) or in post-training (Zhang et al., 2024b) stage, can improve its skill at reasoning tasks, even ones that are unrelated to programming. These prior works have hypothesized that the properties of code data, such as its logical consistency, compositional structure, and reduced ambiguity compared to natural language, provide effective signals that benefit reasoning. Despite the broad effectiveness of code data in training, we still lack a systematic, causal understanding of which specific properties of code drive these reasoning improvements: is it its syntactic regularity, structural abstractions, or linguistic styles?

In this work, we systematically investigate which aspects of code provide effective training signals. To this end, we construct parallel instruction datasets in both natural language and code, and further expand the code dataset into language-specific variants by generating responses in ten widely used programming languages. This design allows us to examine how structural differences across languages affect downstream reasoning. In addition, we introduce controlled perturbations to the code data to isolate contributing factors: (1) *rule-based* transformations such as whitespace removal or comment shuffling, and (2) *generative* transformations where GPT-4o-mini rewrites or reformats the code (e.g., with augmented comments, pseudocode, or flowcharts). We then fine-tune language models on each dataset variant, and evaluate them across natural language and general knowledge, math, as well as code understanding and generation tasks. Our contributions are:

---

[*]Equal contribution

- We introduce a systematic framework to disentangle what aspects of code data improve reasoning, combining parallel instruction data construction, controlled perturbations, and large-scale evaluation across five model families and eight scales.
- We design a comprehensive and controlled suite of perturbations spanning rule-based edits and generative rewritings.
- We provide new insights into the role of code in reasoning to inspire guidance on leveraging its structural and linguistic properties in future training data design.

## 2  RELATED WORK

**Code data for LLM reasoning**  Recent work has increasingly demonstrated that incorporating code data can substantially improve the reasoning abilities of LLMs. Prior studies show that adding code during pre-training or instruction-tuning consistently improves model performance across reasoning tasks, domains, model scales and architectures (Ma et al., 2023a; Zhang et al., 2024a; Yang et al., 2025b; Aryabumi et al., 2024). Several works further explore the synergy between code and reasoning and highlight how code's structured and verifiable properties support logical decomposition and intermediate step generation (Bi et al.; Yang et al., 2024). This effect has been observed in multilingual contexts as well, where code-augmented training improves structured reasoning in under-resourced languages (Li et al., 2024). Complementary research focuses on code's impact for alignment and reward modeling, where pre-training with code-preference pairs or code-based intermediate steps can improve model calibration for reasoning-intensive tasks (Yu et al., 2024). The closest line of research to our work explores stress-testing LLMs with structural and semantic code perturbations (Lam et al., 2025), which shows that small corruptions can significantly reduce reasoning performance.

**Data impact on LLM performance**  The performance of LLMs is tied to the vast amounts of training data, but the quality, composition, and characteristics of this data greatly shape their abilities (Wang et al., 2024; Li et al., 2023; Lee et al., 2022). For example, extensive analyses by Longpre et al. (2024) have shown that pre-training data curation decisions for dataset age, composition, and content filtering have a systematic impact on downstream performance. These effects persist even after fine-tuning steps. Zhang et al. (2024c) demonstrate that poisoning as little as 0.1% (and even 0.001%) can produce persistent behavioral changes that survive instruction-tuning and alignment. In addition, Havrilla & Iyer (2024) showed that LLMs are sensitive to global, accumulative errors in chain-of-thought-structured training data, and that it is critical to filter out documents containing large amounts of dynamic, global noise during both pre-training and fine-tuning .

## 3  METHODOLOGY

We design a controlled experimental framework to understand what aspects of code improve reasoning in language models. Our methodology consists of three stages: constructing parallel natural language and code instruction datasets (§3.1); applying systematic modifications to code instruction data (§3.2); and fine-tuning various language models on each dataset variant and then conducting evaluation (§3.3). An overview of this framework is shown in Figure 1.

### 3.1  INSTRUCTION DATA GENERATION

We construct two parallel instruction datasets: one in natural language and the other in code, each containing 120,000 instruction-response pairs. We collect instructions from publicly available datasets, carefully process and filter them through deduplication and language-agnostic filtering, and augment the code data in a controlled way. This construction enables a more controlled comparison of natural- and code-based instruction following under a unified training framework.

**Code instructions**  We aggregate code instructions from Codeforces-CoT (Penedo et al., 2025), Code-Instruction-122K (TokenBender, 2024), Evol-Instruct-Code-80k-v1 (nickrosh, 2024), Code-Instruction (red1xe, 2023), Code-Instruct-Sets (AtlasUnified, 2023), and Code-Instruct-Alpaca-Vicuna-WizardLM (rombodawg, 2024). We aim to construct instruction data that is high-quality, diverse, and language-agnostic.

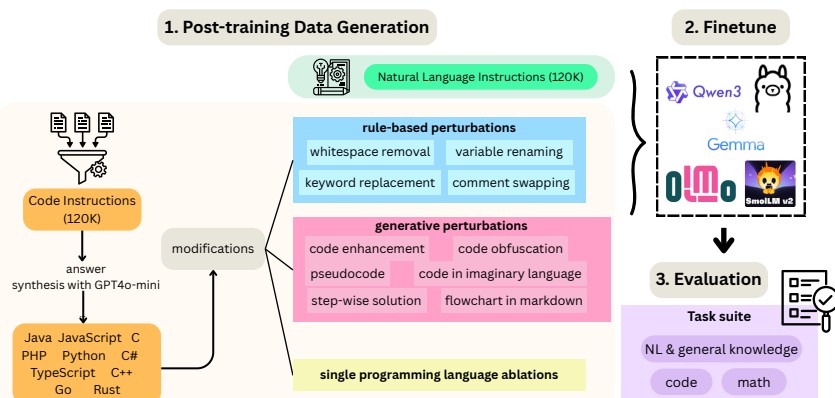

Figure 1: We construct parallel code and natural language instruction datasets, apply targeted modifications (rule-based and generative-based perturbations, single programming language ablations), and fine-tune a separate LLM on each modified dataset. We then evaluate the resulting models across general natural language, code, and math reasoning tasks.

To ensure generality and eliminate redundancy, we first remove all exact-match duplicates across the datasets. We then filter out instructions that are explicitly programming-language-specific (e.g., "Translate this code from Python to java") or whose solutions are inherently tied to particular domains, such as web development or databases (e.g., "webpage", "website", "SQL", "HTML").

For each instruction, we prompt GPT-4o-mini* to generate answers in ten widely used programming languages: Java, JavaScript, PHP, Python, C#, TypeScript, C, C++, Go, and Rust. To create these variants, we design 20 language specification templates that explicitly request a solution in a given programming language (Table 6). For every instruction, we randomly select a template, format it with one of the target languages, and combine it with the general generation instructions to construct a complete prompt (Figure 5). From these generations, we sample 120K instruction–response pairs with valid outputs, evenly distributed across all ten languages.

To assess the quality of our synthesized code instruction–response pairs, we perform a comprehensive syntax and compilation check across all ten programming languages. For each instance, we extract the generated code block and apply standard syntax or compilation tools (e.g., `ast.parse` for Python, `gcc -fsyntax-only` for C, `javac` for Java). As shown in Appendix Table 3, the majority of samples compile or execute successfully, with pass rates ranging from 64.08% (TypeScript) to 99.25% (Python) and an average pass rate of 82.59% across all languages. These results show that most generated instructions correspond to syntactically valid and executable code.

**Natural language instructions** We sample 120K examples from the OpenHermes 2.5 corpus (Teknium, 2023). We exclude instruction-response pairs associated with categories unrelated to general-purpose instruction following, such as "agent" and "summarization", as well as those labeled "coding" to ensure the dataset is entirely natural language. To maintain linguistic consistency, we further filter out non-English examples. This filtered natural language subset complements our code instruction data, enabling a fair comparison between code and natural language instructions.

## 3.2 SYSTEMATIC PERTURBATION DESIGN

To understand which specific structural and semantic properties are responsible for changes in reasoning task performances, we systematically perturb different aspects of the code dataset. We design the perturbations in two ways: *rule-based* (deterministic transformations) and *generative* (model-generated augmentations). Notably, our perturbation strategies do not alter the number of examples in the dataset. We illustrate examples of these perturbations in Table 1, with extended examples and token statistics in Appendix A.1.

---

*Responses are generated with temperature 0.6 and API-default decoding parameters.

Table 1: An example of perturbations (Section 3.2) applied to the same original snippet.

| Full Original Snippet | Type | Strategy | Original Excerpt | Perturbed Excerpt |
|---|---|---|---|---|
| `def process_string(input_string):`
`  vowels = "aoyeuiAOYEUI"`
`  result = []`

`  for char in input_string:`
`    if char not in vowels:`
`      result.append('.' +`
`char.lower())`

`  return ''.join(result)`

`# Read input`
`input_string = input().strip()`
`# Process and print the result`
`print(process_string(input_string))` | Rule-based | Whitespace Removal | `result.append('.' +`
`char.lower())` | `result.append('.'+char.lower())` |
| | | Variable Renaming | `for char in`
`input_string: ...` | `for var_4 in var_1: if var_4`
`not in var_2: ...` |
| | | Keyword Replacement (Nonsense) | `if char not in`
`vowels:` | `garply i not in baz` |
| | | Keyword Replacement (Non-English) | `for char in`
`input_string:` | `para ch en entrada` |
| | | Comment Swapping (Local) | `# Read input` | `# Walking` |
| | | Comment Swapping (Global) | `# Process and print`
`the result` | `// Queue for processing`
`nodes` |
| | | Comment Removal | `# Read input` | `/* all comments removed */` |
| | Generative | Pseudocode | `for char in`
`input_string: if`
`char not in vowels` | *FOR EACH character IF not vowel THEN*
*append '.'+lowercase* |
| | | Step-by-Step | `result.append('.' +`
`char.lower())` | *Append '.' before consonants and convert*
*to lowercase* |
| | | Flowchart | `if char not in`
`vowels:` | *[Read char] → {Vowel?} →*
*[Append '.'+lower]* |
| | | Code in Imaginary Language | `result.append('.' +`
`char.lower())` | *glorf add '.' ⊕ lower(chr)* |
| | | Comment Enhancement | `# Process and print`
`the result` | *# Removes vowels and prefixes consonants*
*with '.'* |
| | | Comment Obfuscation | `# Read input` | *# WARNING: Code may summon aliens; #*
*TODO: handle quantum vowels* |

### 3.2.1 RULE-BASED PERTURBATIONS

Rule-based perturbations apply deterministic transformations to the code. They are designed to disrupt superficial patterns or semantic signals that may influence model predictions without altering the core logic of the code. We describe five such perturbations below:

**Whitespace removal** All whitespace characters are removed from the code. This tests whether models rely on formatting heuristics, such as indentation or visual grouping of blocks, as implicit structural cues, particularly in languages like Python where whitespace is semantically meaningful.

**Variable renaming** We replace user-defined variables, function names, and class names with canonical placeholders of the form `var_i`, where $i \in [0, n)$ and $n$ is the total number of unique identifiers in the code snippet. This removes semantic cues conveyed by meaningful identifier names (e.g., `counter`, `isSorted`).

**Programming language keyword replacement** For each of the ten programming languages in our dataset, we identify its reserved keywords (e.g., `if`, `return`, `def` in Python) and substitute all occurrences of them using two strategies. The first replaces keywords with nonsense tokens (e.g., *foo*, *quux*), which have no semantic meaning in any language. In the second strategy, we use non-English but valid words (e.g., *amigo*, *fleur*), which are real words in various languages but semantically unrelated to the programming context. These perturbations aim to challenge models' reliance on syntactic and semantic cues from familiar language constructs.

**Comment removal** We remove all inline and block comments from each code snippet. Code comments often provide useful semantic signals for program comprehension (Buse & Weimer, 2009; De Souza et al., 2005). This perturbation tests whether models largely leverage such auxiliary natural-language cues.

**Comment swapping** We introduce local and global swapping that misplace code comments to disrupt the semantic alignment between code and documentation. In local swapping, comments within a snippet are randomly reordered, preserving their content but misaligning them with the relevant code segments. In global swapping, we first collect a global pool of comments from the entire dataset. Then, for each comment in a snippet, we replace it with a randomly sampled comment from this pool. This results in documentation that is entirely mismatched to the surrounding code.

### 3.2.2 GENERATIVE PERTURBATIONS

We create generative perturbations by prompting GPT-4o-mini[†] to produce alternative versions of code responses generated according to Section 3.1. These rewrites preserve the original intent of the code while introducing more diverse variations beyond what rule-based edits can achieve, allowing us to test model sensitivity and robustness to semantically equivalent inputs expressed in different forms. The full set of prompts used is available in Appendix A.6.

**Comment enhancement** We prompt GPT-4o-mini to regenerate the code with high-quality documentation and inline comments (Figure 6). The prompt emphasizes two forms of annotation: (1) comprehensive documentation comments for all functions, classes, and key code blocks to describe their purpose, parameters, return values, and assumptions; and (2) informative inline comments that clarify complex or non-obvious logic. These annotations follow the conventions of the target programming language (e.g., Python docstrings, JavaDoc). Unlike the often sparse comments in unperturbed data, the enhanced versions provide consistent, high-quality annotations, which enable us to test the effect of documentation on model performance.

**Comment obfuscation** Here, we generate deliberately misleading, irrelevant, or nonsensical comments, while preserving the code's functionality (Figure 7). These include (1) inaccurate, off-topic, or absurd documentation (e.g., references to astrology, cooking, or fictitious technologies) and (2) chaotic inline comments that contradict the code's functionality, reference imaginary bugs or features, and use distracting styles such as ALL-CAPS, emojis, and fabricated jargon. This perturbation tests model robustness to extreme noise and deceptive annotations.

**Pseudocode** We convert code into high-level pseudocode while preserving its logical structure (Figure 8). The model is instructed to replace language-specific syntax with pseudocode constructs (e.g., `IF...THEN...ENDIF`, `FOR EACH`, etc.), remove low-level implementation details (e.g., type declarations or library calls), and maintain the original control flow and indentation. This perturbation evaluates whether models can reason over algorithmic intent without relying on concrete syntax, which offers insight into generalization across abstraction layers in code representation.

**Flowchart in Markdown** We generate a control flow diagram using Mermaid syntax in Markdown for a given code snippet (Figure 9). The diagram captures all major control structures, such as loops, branches, function calls, and return points, using minimal but descriptive labels. This transformation renders executable code as a graphical abstraction, allowing us to understand whether models can reason over symbolic control flow and align it with underlying program semantics.

**Step-by-step solution** We rewrite code as a numbered list of natural language steps (Figure 10). Each step preserves the program's logic and execution order but uses declarative, language-agnostic phrasing (e.g., "Define a function named...", "Check if the input is valid"). Unlike pseudocode or flowchart formats, this version entirely removes code or symbolic notation and instead emphasizes procedural understanding in purely narrative form.

**Code in imaginary language** We translate real code into a fictional language that preserves structure and control flow but replaces all syntax and identifiers with invented tokens (Figure 11). The result is semantically consistent yet entirely ungrounded in real languages. This perturbation allows us to examine whether models rely on surface-form familiarity (e.g., recognizing logical patterns) rather than true program semantics.

To assess the correctness of the perturbed data, we conduct a human evaluation with two annotators, randomly sampling 30 examples per perturbation type (13 total: 7 rule-based and 6 generative). For the rule-based perturbations and comment enhancement/obfuscation, annotators verify that each transformation strictly follows the intended perturbation rule while leaving all unrelated content unchanged. For the generative perturbations (pseudocode, step-by-step instructions, flowchart, imaginary language), which express the original code in alternative forms, annotators verify that the conveyed semantics remain faithful to the original program. Across all 390 sampled instances, 351 were judged correct (90% overall). Rule-based perturbations achieved $176/210 \approx 84\%$ correctness, while generative perturbations achieved $175/180 \approx 97\%$ correctness.

---

[†]We use temperature of 0.6 and default settings.

## 3.3 MODEL TRAINING AND EVALUATION

We train a suite of decoder-only LLMs using supervised fine-tuning (SFT) on our instruction–response datasets as detailed in Section 3.1, along with their perturbed variants described in Section 3.2. To assess the effect of language-specific patterns, we additionally fine-tune models on subsets of the code data restricted to a single programming language. This allows us to examine how the syntactical diversity of programming languages influences reasoning performance. Each instruction–response pair is treated as a single input–output sequence, and models are trained to autoregressively predict the response tokens conditioned on the instruction and prior context. All models are fine-tuned from the same pre-trained backbone to ensure comparability across experimental conditions. Let $x = (x_1, x_2, \ldots, x_m)$ be the instruction tokens and $y = (y_1, y_2, \ldots, y_n)$ be the response tokens. The SFT objective is defined as:

$$\mathcal{L}_{\text{SFT}} = -\sum_{t=1}^{n} \log P_\theta(y_t \mid x, y_{<t}) \tag{1}$$

where $P_\theta$ denotes the model's conditional probability distribution parameterized by $\theta$, and $y_{<t}$ represents the prefix of the response up to position $t - 1$.

**Models**  We choose a diverse set of pre- and post-trained language models ranging from 0.6B to 8B parameters. Specifically, we experiment with models from five major families: Qwen3 (Yang et al., 2025a), LLaMA-3 (Grattafiori et al., 2024), Gemma3 (Team et al., 2025), OLMo2 (OLMo et al., 2024), and SmolLM2 (Allal et al., 2025). For each model family, we select representative sizes (e.g., $<$1B, $\sim$1B, $\sim$3-4B, $\sim$7-8B)[‡] to evaluate performance across different scales.

**Training data configurations**  Our base training set consists of 120K instruction–response pairs spanning both code and natural language formats as detailed in Section 3.1. From this, we construct several configurations: (1) 100% code-only, (2) 100% natural language-only, and (3) mixed data with varying code-to-language ratios. In addition, we train models on each perturbed variant introduced in Section 3.2. Finally, we include programming-language-specific subsets, training separate models on data from each of the ten languages ($\sim$12K examples per language) to assess the effect of language specialization. The implementation details are in Section A.5.

**Evaluation tasks**  We evaluate model performance across three categories: natural language and general knowledge, math, and code (Table 4).

For natural language and general knowledge, we evaluate across commonsense reasoning, science, and textbook-style QA, logical reasoning, and instruction-following. All tasks are evaluated using accuracy. For math, we include both elementary and advanced problem-solving datasets (e.g., GSM8K, HRM8K), as well as arithmetic and math-related subsets of MMLU. Open-ended tasks (GSM8K, HRM8K) use exact match, while arithmetic and MMLU (math) are scored with accuracy.

For code, we evaluate both code understanding and generation. Based on preliminary experiments, we adopt the LLM-as-Judge paradigm (Zheng et al., 2023a; Gu et al., 2025) instead of execution-based evaluation (Huang et al., 2022). Our relatively small, perturbed models often fail to produce fully executable code, making execution-based metrics unreliable. More importantly, our goal is to assess code quality and reasoning under perturbations, not just execution success.

Thus, we prompt *GPT-4o-mini* to first generate an instance-specific rubric on a 1–10 Likert-type scale given the original instruction, which is expected to capture nuanced quality variation across outputs. The same model is then prompted as a judge to provide a brief reasoning step ("thought") and assign a score based on that rubric. Examples of the rubric-generation prompt and judging prompt are shown in Appendix A.6 (Figures 12 and 13). For the main results, we use *GPT-4o-mini* as the judge due to its strong judging quality and favorable cost–performance tradeoff. To assess the reliability of our LLM-as-judge setup, we additionally conduct an extensive cross-judge analysis using multiple models. The results in Appendix Table 7 demonstrate that our evaluation is stable across judges.

---

[‡]Due to resource constraint, the largest model we could fully finetune is 8B.

## 4 RESULTS AND DISCUSSION

**RQ1: Does incorporating code in fine-tuning improve task performance?** First, we validate prior findings that fine-tuning on code data can enhance reasoning. Following the training setup in Section 3.3, we compare performance across four settings: zero-shot, full code fine-tuning ("code-ft"), full natural language fine-tuning ("nl-ft"), and mixed data fine-tuning with equal proportions of code and natural language instructions ("mixed-ft"). Across model families and scales, code-ft and mixed-ft generally achieve leading or competitive performance across tasks (Figure 2, and Figures 14–18), with the trend particularly consistent on code generation.

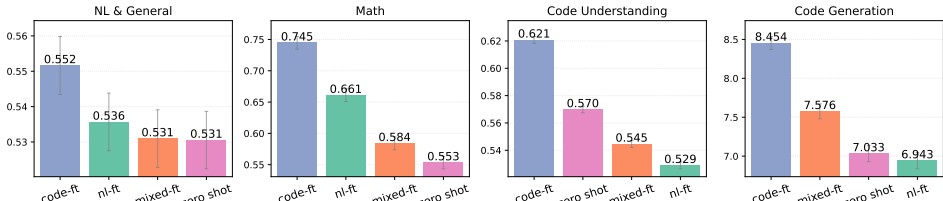

Figure 2: Performance (with stderr bars) of Qwen3-4B-Base across zero-shot, full code fine-tuning (code-ft), full natural language fine-tuning (nl-ft), and 50-50 code to NL data ratio fine-tuning (mixed ft). Incorporating code improves performance across tasks.

Overall, across the 14 model bases, either code-ft or mixed-ft achieves the best performance on 64% of natural language tasks, 86% of math and code understanding tasks, and all code generation tasks. Motivated by this, we further examine the effect of varying the proportion of code in mixed fine-tuning (Figure 19). We find that higher fractions of code data generally improve performance across most tasks, with math tasks most sensitive to mixture ratios.

**RQ2: How do our systematic perturbations affect performance?**

> **Section Findings**
>
> - Structural perturbations hurt more than semantic ones, especially for math and code.
> - Appropriate abstractions, such as pseudocode and flowcharts, can substitute for explicit code structure in reasoning.
> - Models don't need verbose code: reduced-token variants perform well as long as core information is preserved.
> - LLMs can reason effectively from corrupted code by exploiting surface-level regularities.

Next, we analyze task performance under the perturbations introduced in Section 3.2. Based on the properties of each perturbation, we group them into distinct analysis axes that allow us to systematically probe their effects. The grouping details are in Table 5. We present the results for individual perturbations in Appendix A.7.6.

**Structural vs. Semantics Perturbations.** We define structural perturbations as edits that alter the syntactic scaffolding or formatting of code (e.g., whitespace removal, pseudocode, flowcharts), while semantic perturbations modify meaning-bearing tokens such as identifiers, keywords, or comments without disrupting the underlying structure. Across model families and scales (Figures 20 – 24), nearly all perturbations reduce performance compared to the unperturbed code fine-tuned baseline. More importantly, structural perturbations consistently degrade performance more severely than semantic ones, especially for math and code tasks (e.g., Figure 3a). The discrepancy is more evident as models scale up (e.g., Figure 20). This resembles prior work that reasoning structure rather than content is more critical to the learning process (Li et al., 2025). We hypothesize that tasks such as math and code rely more heavily on formatting and layout cues to shape reasoning.

**Explicitness of Code Structure.** Building on the importance of structure, we examine perturbations along a spectrum of how explicitly they preserve code structure: from runnable or code-like forms, through intermediate abstractions such as pseudocode and flowcharts, to natural language

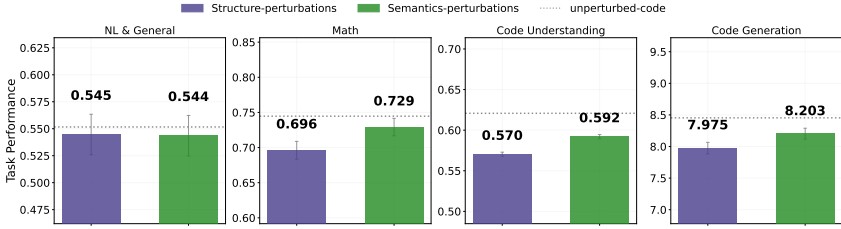

(a) Structural (e.g. removing whitespace) vs. semantics perturbations (e.g. modifying the comments) of Qwen3-4B-Base. Structural perturbations tend to be more harmful to performance than semantic ones.

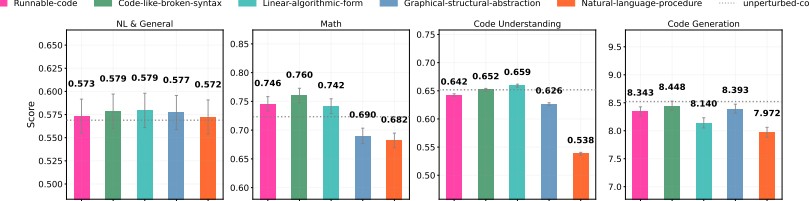

(b) Performance under levels of explicitness of code structure (less explicit going from runnable code to NL procedure) of Qwen3-8B-Base. Certain algorithmic and graphical abstractions benefit reasoning.

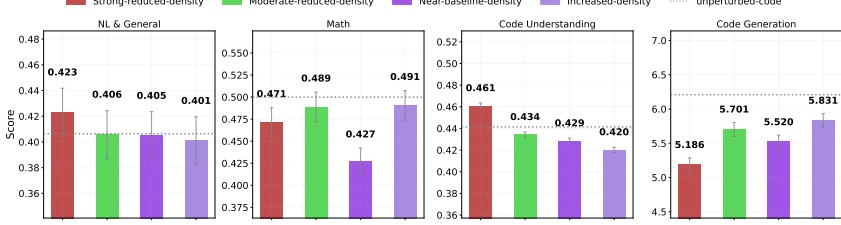

(c) Performance under token count reductions w.r.t. unperturbed code (Qwen3-0.6B-Base). Reductions can perform comparable or even better than the baseline.

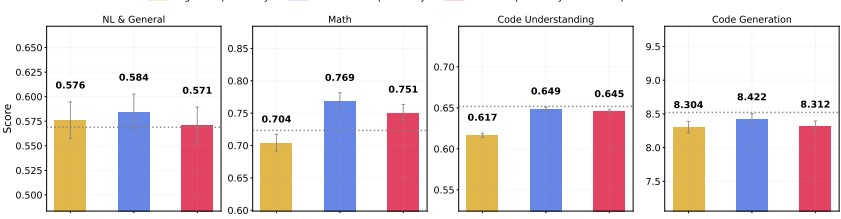

(d) Performance vs. human interpretability of perturbed code (Qwen3-8B-Base). Low-interpretability with misleading signals can match or perform better than other configurations.

Figure 3: Aggregated performance (with stderr bars) across RQ2 perturbation settings: structural vs. semantic perturbations, explicitness of code structure, token density reductions, and human interpretability.

step-by-step procedures. For code generation, where executable outputs are required, it is natural that perturbations that preserve explicit code structure, whether runnable or not, lead to the best performance. For other tasks, however, certain abstractions such as pseudocode or flowcharts often match or even surpass unperturbed code, as they highlight algorithmic structure while removing superficial syntax. By contrast, the most implicit form, natural language procedures, provides little advantage and generally performs worst across tasks (e.g. Figure 3b, Figures 25–29).

**Relative Information Density.** Because our constructed instruction datasets are parallel, the amount of information they convey about the code is comparable across perturbations. We define relative information density as (number of tokens in perturbed dataset) ÷ (number of tokens in the original

code-ft dataset), which reflects how compactly the same content is represented. Perturbations differ in how they adjust density: highly compact forms strip away most tokens while preserving the algorithmic skeleton (e.g., flowcharts, pseudocode), moderately reduced forms remove comments or use imaginary languages, and others preserve or even increase density through verbose variable renamings or enriched documentation. We find that strong or moderate reductions in density often perform close to, and sometimes better than, the baseline (e.g., Figure 3c, Figures 30–34). However, this advantage doesn't extend to code generation, where preserving richer surface detail is important. In addition, smaller models are more sensitive to density differences, whereas larger models remain robust. Overall, this suggests that the benefit of code for reasoning doesn't lie in its verbosity but in the efficiency with which essential information is preserved.

**Human Interpretability.** We also examine perturbations through the lens of human readability: high-interpretability (enriched explanations and visual scaffolds), medium (local edits leaving most code intact), and low (obscured readability or misleading signals). Interestingly, low-interpretability variants, despite adding noise or distortion, often do not degrade performance too much from the unperturbed baseline, and often match or even surpass medium-interpretability ones (e.g. Figure 3d, Figures 35–39). This counterintuitive trend suggests that the models could exploit surface-level regularities and recurring structural cues that persist even in noisy or opaque forms.

**RQ3: How does performance vary across programming languages?**

> **Section Findings**
> - Lower-level languages benefit math tasks.
> - Python aligns best with NL tasks, while Java and Rust often rank among the top for math.

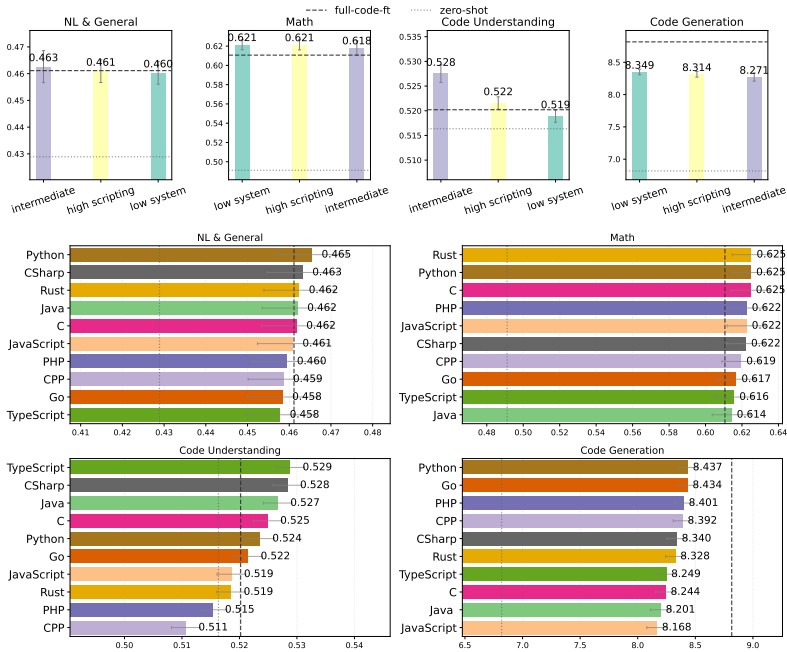

Figure 4: Performance (with stderr bars) of Qwen3-1.7B. *Top:* grouped by abstraction level (low-system, intermediate, high-scripting). Low-system and intermediate languages outperform on math. *Bottom:* individual programming languages. Python aligns best with NL, Rust leads on math.

The impact of structure in RQ2 motivates the question of whether syntactic regularities in programming languages also influence model performance. To explore this, we group the ten programming languages into high-scripting (Python, PHP, JavaScript, TypeScript), intermediate (Java, C#), and low-system (C, C++, Rust, Go) according to their abstraction level. Overall, differences across groups are small. On NL and code tasks, the impact of language groups is largely model-dependent. However, on math tasks, most high-scripting languages consistently underperform relative to in-

termediate and low-system ones (e.g. top Figure 4, Figures 45–48a). We hypothesize that richer structural detail in lower-level languages provides beneficial signals for mathematical reasoning.

For code generation, fine-tuning on any single language improves over zero-shot but lags behind full code fine-tuning, which suggests the benefit of multi-language diversity for code generation. At the individual language level (e.g. bottom Figure 4, Figures 46–48b), across models, Python often leads on NL tasks, probably due to its surface form being closer to natural language. Aligning with the group-level results, lower-level languages such as Java and Rust often rank among the top for math. For code tasks that span multiple languages, results are more mixed, with no clear leaders, and performance gaps remain relatively small.

## 5 Conclusion

In this work, we investigate what aspects of code enhance reasoning in LLMs and which factors matter most. Through 3,331 experiments spanning five model families, eight scales, ten programming languages, and a suite of systematic perturbations, we arrive at four central conclusions. First, structural properties of code are critical: disrupting them leads to consistent performance drops, especially on math and code tasks. Second, appropriate abstractions and efficient encodings can be just as effective as raw code. Moreover, models remain surprisingly robust even to corrupted or low-interpretability code, exploiting regularities that persist despite surface distortions. Finally, lower-level programming languages provide more benefits for math tasks. Together, we provide a more precise account of how code supports reasoning and point toward practical design principles for constructing effective training data beyond executable programs.

## 6 Limitations and Future Work

Our study focuses on small- to mid-size base models due to resource constraints. Our perturbations, although diverse, do not exhaustively cover all factors that may influence reasoning, such as code complexity or training data diversity. While we do perform manual inspection of a small subset of perturbed examples to verify that transformations behaved as intended, the scale of our dataset precluded a comprehensive human evaluation. Finally, although we evaluate across a broad suite of reasoning tasks, our benchmarks still capture only part of the reasoning spectrum.

Future work could extend our framework to larger and instruction-tuned models to examine whether the trends we observe persist at scale. Exploring additional domains such as chain-of-thought reasoning and preference optimization would further clarify the mechanisms by which code shapes reasoning. More broadly, our findings on abstraction level and information density point toward practical guidelines for training data curation that could be operationalized into automated data selection pipelines.

## 7 Reproducibility Statement

We provide extensive details throughout the paper and supplementary materials. Section 3.1 describes the construction and processing of both the code and natural language datasets. Section A.5 outlines model training and implementation details. Appendix A.6 includes all prompts used for data generation, perturbations, and LLM-as-Judge evaluation.

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

# A APPENDIX

## A.1 EXTENDED DETAILS OF PERTURBATION DATA

For each perturbation, we report the original and transformed code along with token statistics in Table 2.

Table 2: Perturbation type along with the corresponding original and transformed excerpts and token statistics, illustrating how different perturbations alter code structure and information density.

| Perturbation | Original Excerpt | Perturbed Excerpt | Total Tokens | Avg Tokens per Instruction |
|---|---|---|---|---|
| whitespace removal | `for char in input_string:` | `forcharininput_string:` | 78,553,430 | 654.61 |
| variable renaming | `for char in input_string:` | `for var_4 in var_1:` | 87,619,500 | 730.16 |
| keyword replaced with non-sense | `if c not in vowels:` | `garply c not in vowels:` | 87,123,587 | 726.03 |
| keyword replaced with non-English | `if c not in vowels:` | `père c not in vowels:` | 88,078,906 | 733.99 |
| comment removal | `# Read input` | `-` | 80,238,050 | 668.65 |
| local comment swapping | `# Read input` | `# Process and print the result` | 85,324,436 | 711.04 |
| global comment swapping | `# Process and print the result` | `// Queue for processing nodes` | 85,329,862 | 711.08 |
| flowchart (Markdown) | `if char not in vowels:` | `[Read char] → {Vowel?} → [Append '.' + lower]` | 67,553,461 | 562.95 |
| step-by-step explanation | `result.append('.' + char.lower())` | `Append '.' before consonants ...` | 84,250,378 | 702.09 |
| pseudocode | `for char in input_string:` | `FOR EACH character IF not vowel THEN` | 73,722,933 | 614.36 |
| imaginary language | `result.append('.' + char.lower())` | `gloff add '.' ⊕ lower(chr)` | 81,011,032 | 675.09 |
| comment enhancement | `# Process the result` | `# Removes vowels and prefixes consonants ...` | 119,399,621 | 994.99 |
| comment obfuscation | `# Read input` | `# WARNING: Code may summon aliens ...` | 111,771,640 | 931.43 |

## A.2 VERIFICATION OF QUALITY OF SYNTHETIC CODE DATA

As shown in Table 3, most samples pass syntax and compilation checks across languages, with an average pass rate of 82.59%. This indicates that our code data is largely syntactically valid.

## A.3 EVALUATION SUITE DETAILS

We provide details about evaluation data in Table 4.

Table 3: Syntax and compilation check results across all ten programming languages. The majority of samples successfully compiled or executed, with a mean pass rate of 82.59%

| Language | Total | % Passed |
|---|---|---|
| C | 11,998 | 81.49 |
| PHP | 12,009 | 94.81 |
| JavaScript | 11,996 | 91.57 |
| Python | 11,993 | 99.25 |
| C++ | 11,997 | 83.20 |
| TypeScript | 12,001 | 64.08 |
| Rust | 11,995 | 66.71 |
| C# | 11,996 | 81.06 |
| Go | 12,012 | 77.77 |
| Java | 12,003 | 88.94 |

Table 4: Evaluation suite spanning natural language and general knowledge, math, and code tasks.

| Task Type | Topic | Benchmarks | Metric |
|---|---|---|---|
| Natural Language & General Knowledge | Commonsense | PIQA (Bisk et al., 2019) | Accuracy |
| | Science / Textbook | ARC-Easy (Clark et al., 2018) ARC-Challenge (Clark et al., 2018) OpenBookQA (Mihaylov et al., 2018) MMLU (non-math) (Hendrycks et al., 2021) | |
| | Logic-Heavy | LogiQA (Liu et al., 2020) | |
| | Instruction Following | IFEval (Zhou et al., 2023) | Prompt-level Accuracy |
| Math | – | GSM8K (Cobbe et al., 2021) HRM8K (Ko et al., 2025) | Exact Match |
| | – | Arithmetic (Brown et al., 2020) MMLU (math) (Hendrycks et al., 2021) | Accuracy |
| Code | Code Understanding | CodeMMLU (Manh et al., 2024) | Accuracy |
| | Code Generation | HumanEvalX (Zheng et al., 2023b) | LLM-as-Judge |

## A.4 Categorization of perturbations for RQ2 analysis

We group all perturbations across four axes: structural vs. semantic changes, explicitness of code structure (ECS), relative information density (RID), and human interpretability (HI), as summarized in Table 5.

## A.5 Implementation details

We train all models under identical hyperparameter settings to ensure a fair comparison across model sizes and data configurations. All experiments are conducted using full fine-tuning in *BF16* precision with a maximum sequence length of 2048 tokens. We run all experiments on $4\times$A100 80G node. Models are trained for 2 epochs with a cumulative batch size of 64 for most experiments, except for language-specific settings, where the batch size is reduced to 32. The learning rate is fixed at $1e-5$ and follows a cosine decay schedule with a warmup ratio of 0.1. For memory-efficient parallelism and distributed training, we use *DeepSpeed ZeRO Stage 3* (Ren et al., 2021). All models are trained using the LLaMA-Factory (Zheng et al., 2024). All other parameters and configurations follow the default setting unless otherwise specified.

## A.6 Prompts

**Standard generation prompt** We provide the standard prompt to generate code for a given instruction in a specific language in Figure 5. , where the *instruction* can be instantiated using one of the templates in Table 6.

Table 5: Categorization of perturbations across four analysis axes: structural vs. semantic (S/S) perturbations, explicitness of code structure (ECS), relative information density (RID), and human interpretability (HI).

| Perturbation | S/S Perturbations | ECS | RID | HI |
|---|---|---|---|---|
| Whitespace removal | | Broken syntax | Moderate-reduced | Medium |
| Pseudocode | | Algorithmic | Strong-reduced | High |
| Imaginary | Structural | Broken syntax | Moderate-reduced | Low |
| Step-by-step | | NL procedure | Moderate-reduced | High |
| Flowchart | | Graphical | Strong-reduced | High |
| Comment removal | | Runnable | Moderate-reduced | Medium |
| Variable renaming | | Runnable | Increased | Medium |
| Keyword repl. (nonsense) | | Broken syntax | Increased | Low |
| Keyword repl. (non-Eng.) | Semantic | Broken syntax | Increased | Low |
| Comment swap (global) | | Runnable | Near-baseline | Low |
| Comment swap (local) | | Runnable | Near-baseline | Low |
| Comment enhancement | | Runnable | Increased | High |
| Comment obfuscation | | Runnable | Increased | Low |

Table 6: Language specification templates with placeholders that can be instantiated with different programming languages.

| | | |
|---|---|---|
| Generate the code in {language}. | Provide code in {language}. | Write the code in {language}. |
| Build the code using {language}. | Create the code using {language}. | Draft the code in {language}. |
| Produce a code snippet in {language}. | Develop the code using {language}. | Generate a solution in {language}. |
| Create a script in {language}. | Implement the code in {language}. | Design the code in {language}. |
| Construct the code using {language}. | Format the code in {language}. | Write a program in {language}. |
| Prepare a code snippet in {language}. | Write a function in {language}. | Deliver the code in {language}. |

**Comment enhancement prompt**   The prompt to enhance the quality and readability of a given code snippet by adding detailed documentation is shown in Figure 6.

**Comment obfuscation prompt**   The prompt used to generate obfuscated versions of code from a given instruction is presented in Figure 7.

**Pseudo generation prompt**   We illustrate the prompt designed to produce pseudocode for a given instruction in Figure 8.

**Flowchart generation prompt**   The prompt for generating a flowchart-style representation of an instruction is provided in Figure 9.

**Step-by-step implementation guide generation prompt**   The prompt used to create a sequential step-by-step implementation guide for an instruction is shown in Figure 10.

**Imaginary language code generation**  We paragraph the prompt for generating code in an imaginary programming language in Figure 11.

**LLM-as-Judge Evaluation**   We use the prompt shown in Figure 12 to generate instance-specific rubrics for LLM-as-judge evaluation on the code generation task. The prompt to evaluate model response is shown in the Figure 13.

---

**Code Instruction Data Generation Prompt**

You are tasked with generating code based on a specified programming language and instruction. Your goal is to generate code that follows the syntax and semantics of the specified language. If the instruction is invalid (e.g., contradicts the language's rules or references functions or constructs from a different language), you must strictly respond with "invalid."

**Guidelines:** - **Valid Code:** - The generated code must be syntactically and semantically correct according to the specified language. - The code should follow standard conventions and best practices for the given language. - Do **not** provide any explanation for valid code — only output the code itself.
- **Invalid Instruction:** - If the instruction references constructs, functions, or syntax not supported by the specified language, respond with '"invalid"'. - Do **not** attempt to correct the invalid instruction — just respond with '"invalid"'. - Do **not** provide a reason or explanation for why the instruction is invalid.

**Examples:**
Example 1:
Instruction: "Write a function to convert a list to a set."
Language: Python
Response:

```
def list_to_set(input_list):
    return set(input_list)
```

Example 2:
Instruction: "Create a class with a method that prints 'Hello' using console.log()."
Language: Python
Response: invalid
Example 3:
Instruction: "List all files, including hidden ones, in the current directory."
Language: Shell
Response: ls -a
Example 4:
Instruction: "Define a function using 'def' that returns the length of a string."
Language: JavaScript
Response: invalid

Instruction:
If the instruction is valid, output the code directly (no explanations).
If the instruction is invalid, respond with "invalid" (no explanation).

**Input:** Instruction: {instruction}
Language: {language}

**Output:**
{{response}}

Figure 5: Code instruction data generation prompt. The task is to generate valid code or respond with "invalid" for unsupported instructions.

---

**Comment Enhancement Prompt**

You are tasked with enhancing the response to the given code instruction by adding meaningful comments and documentation. The goal is to improve the code's readability, maintainability, and clarity across any programming language, without altering its original logic or structure.
**Your modifications must include:**
1. **Documentation Comments**: - Add clear, technically accurate, and concise documentation for every function, method, class, and major code block. - Describe the purpose, all parameters (with correct types and usage), return values, and any assumptions or notes relevant to correct usage. - Use the standard documentation format appropriate for the programming language (e.g., Python docstrings, JavaDoc for Java, Doxygen for C/C++).
2. **Inline Comments**: - Insert informative and contextually helpful inline comments near complex, unintuitive, or important operations. - Focus on explaining logic, control flow, edge-case handling, design decisions, or dependencies. - Avoid redundant, obvious, or overly literal comments (e.g., avoid "i = 0 // set i to 0").
**Guidelines:**
- **Do not change the logic, structure, or behavior** of the original code. - Do not introduce new functionality, abstractions, or formatting changes. - Keep comments strictly technical, relevant, and useful—avoid verbosity or informal tone. - Do not include any meta-comments or explanatory notes about what was changed (e.g., no "This version adds comments" or similar). - Apply these modifications **only** to the code portion of the original response. Leave any non-code parts completely unmodified. - Ensure all comments follow the style conventions of the language in use.
This task is language-agnostic and applies to any of the following: Python, Java, C++, JavaScript, PHP, TypeScript, Go, Rust, C#, Ruby, Swift, Kotlin, Scala, and Shell.
**Input:** Instruction: {instruction} Original Response: {response}
**Output:** {{updated response}}

Figure 6: Comment enhancement prompt. The task is to improve code clarity through meaningful comments while preserving original functionality.

## A.7 EXTENDED RESULTS

### A.7.1 TASK PERFORMANCE SHOWCASING CODE DATA IMPACT IN FINETUNING (RQ1)

**Qwen3 model family results** See task performance of zero-shot, full code finetuned, full natural language finetuned, and code-NL mixed finetuned models in Figure 14.

---

**Comment Obfuscation Prompt**

**You are tasked with modifying the response to the given code instruction in a way that significantly degrades its quality and clarity.** Your goal is to make the code as **confusing, unhelpful, and misleading** as possible through the use of deliberately bad, distracting, and nonsensical comments—without altering the actual logic or functionality of the code.
**Your modifications must include:**
1. **Severely Misleading or Irrelevant Documentation Comments**: - Add documentation to every function, class, and major code block that includes wildly inaccurate descriptions, unrelated facts, inside jokes, cryptic advice, or philosophical ramblings. - Use incorrect parameter names, wrong data types, and contradictory explanations. - Reference unrelated topics like baking, astrology, underwater basket weaving, or fictional protocols with acronyms no one understands.
2. **Chaotic Inline Comments**: - Insert inline comments that blatantly contradict the actual functionality of the code. - Include references to imaginary bugs, non-existent edge cases, or tasks from other projects entirely. - Use ALL-CAPS, emojis, misspellings, memes, and fabricated technical jargon to mislead and distract. - Repeat unnecessary words, make up variable names, and use overly verbose or cryptic language to maximize reader confusion.
**Guidelines:**
- **Do not modify the actual logic, syntax, or structure of the code** — only the comments must be altered. - All comments must remain syntactically valid for the language (e.g., use # for Python, // for JavaScript, etc.) so the code can still execute normally. - Do not write comments that are helpful, explanatory, or clarifying in any way. Remove any useful comments that were originally present. - Do not include any reflective or meta statements about the task (e.g., no "this version degrades the comments"). - Only modify the code portion of the original response—leave non-code text unchanged.
This task is language-agnostic and applies to any of the following: Python, Java, C++, JavaScript, PHP, TypeScript, Go, Rust, C#, Ruby, Swift, Kotlin, Scala, and Shell.
**Input:** Instruction: {instruction} Original Response: {response}
**Output:** {{updated response}}

Figure 7: Comment obfuscation prompt. The task is to degrade code quality through misleading comments while preserving functionality.

---

**Pseudocode Conversion Prompt**

**You are tasked with converting a given code response into pseudocode that mirrors the structure and semantics of the original code, while preserving the idiomatic style of the original programming language.**
**Your modifications must include:**
1. **Pseudocode Style**: - Replace exact syntax with **language-specific pseudocode** constructs (e.g., use IF ... THEN ... ENDIF for conditionals, FOR EACH or WHILE for loops). - Remove implementation details such as variable declarations with types, precise syntax, or specific library calls—replace them with clear, high-level descriptions.
2. **Structure Preservation**: - Maintain the **overall control flow and indentation** of the original code. - Use **meaningful, readable names** that reflect their purpose in the code. - Ensure each function, class, or logical block is represented clearly in pseudocode format.
3. **Fidelity to Language Idioms**: - Adapt the pseudocode to **reflect the spirit and conventions** of the original language (e.g., Python's indentation style, Java's block structure, C++-like modularity).
**Guidelines:**
- **Do not alter the logic, structure, or order** of operations. - **Do not include actual code syntax** (e.g., semicolons, colons, type annotations, brackets). - **Do not add comments, explanations, or headings** outside the code block. - Output only the converted pseudocode. - Preserve formatting and indentation faithfully.
**Input:** Instruction: {instruction} Original Response: {response}
**Output:**

```
{{pseudocode}}
```

Figure 8: Pseudocode conversion prompt. The task is to translate real code into structured pseudocode while preserving logic and idiomatic style.

**Llama-3.2 model family results**    See task performance of zero-shot, full code finetuned, full natural language finetuned, and code-NL mixed finetuned models in Figure 15.

**Gemma-3 model family results**    See task performance of zero-shot, full code finetuned, full natural language finetuned, and code-NL mixed finetuned models in Figure 16.

**OLMo-2 model family results**    See task performance of zero-shot, full code finetuned, full natural language finetuned, and code-NL mixed finetuned models in Figure 17.

**SmolLM2 model family results**    See task performance of zero-shot, full code finetuned, full natural language finetuned, and code-NL mixed finetuned models in Figure 18.

**Code data mixture ratio in fine-tuning data ablations**    We show results for mixing different ratios of code data in fine-tuning for Qwen3-0.6B-Base and Qwen3-1.7B-Base in Figure 19a and Figure 19b, respectively.

---

**Flowchart Generation Prompt**

**You are tasked with generating a flow diagram in Markdown format that visualizes the control flow of the given code response.** Your output must be a **Mermaid** flowchart embedded in a single fenced code block.
**Your diagram must:**
1. **Translate code logic into control flow**: - Include major steps, function calls, loops, branches, and return points. - Use concise, descriptive node labels that accurately reflect the code behavior.
2. **Follow valid Mermaid syntax**: - Begin with `Start` and end with `End`. - Use `[ ]` for actions/processes. - Use `{ }` for decision/branch points. - Use `-->` to connect nodes. - Wrap everything in triple backticks with `mermaid` specified.
3. **Respect language conventions**: - Match naming and idioms to the language used in the original code. - Do not reinterpret or alter the code logic.
**Guidelines:**
- **Do not change the structure or logic** of the original response. - **Do not generate new code**, only a flowchart of the existing response. - Keep node labels technical and minimal. - Do **not** include explanations, comments, or narrative outside the flowchart. - Follow the same formatting and structural conventions as the original prompt.
**Input:** Instruction: {instruction} Original Response: {response}
**Output:**

```
```mermaid
{{flowchart}}
```
```

Figure 9: Flowchart generation prompt. The task is to convert real code into a Mermaid flow diagram without changing logic or structure.

---

**Step-by-Step Generation Prompt**

**You are tasked with converting a given code response into a step-by-step implementation guide that describes how to manually implement the code in clear, concise, and technically accurate language.**
**Your implementation guide must:**
1. **Preserve Original Logic**: - Follow the same structure, logic, and sequence as the original code. - Include all major steps, control structures, and decisions.
2. **Describe, Don't Translate**: - Do not include code or pseudocode. - Write in declarative, instructional sentences that explain what to do and how to do it. - Use neutral, language-agnostic terminology (e.g., "Define a function named...", "Check if...", "Return the result...").
3. **Be Clear and Concise**: - Number each step in the order it occurs. - Use precise and unambiguous language. - Each step should focus on a single coherent action.
**Guidelines:**
- **Do not add extra commentary, examples, or assumptions**. - **Do not change the original logic or execution order**. - **Do not output anything other than the numbered steps**. - Output the guide as a plaintext numbered list only—no code blocks, no explanations outside the list.
**Input:** Instruction: {instruction} Original Response: {response}
**Output:**

```
1. {{Step one}}
2. {{Step two}}
3. {{Step three}}
...
```

Figure 10: Step-by-step implementation guide prompt. The task is to describe how to implement the code in a precise, ordered, and language-agnostic way.

### A.7.2 TASK PERFORMANCE UNDER PERTURBATIONS AGGREGATED BY STRUCTURE VS SEMANTICS (RQ2)

**Qwen3 model family results (structure vs semantics perturbations)** See performance of aggregated task performance under structure vs semantics perturbations in Figure 20.

**Llama-3.2 model family results (structure vs semantics perturbations)** See performance of aggregated task performance under structure vs semantics perturbations in Figure 21.

**Gemma-3 model family results (structure vs semantics perturbations)** See performance of aggregated task performance under structure vs semantics perturbations in Figure 22.

**OlMo-2 model family results (structure vs semantics perturbations)** See performance of aggregated task performance under structure vs semantics perturbations in Figure 23.

**SmolLM2 model family results (structure vs semantics perturbations)** See performance of aggregated task performance under structure vs semantics perturbations in Figure 24.

---

**Imaginary Language Translation Prompt**

You are tasked with converting a given code response into an imaginary programming language that mimics the syntax and semantics of the original real-world language while appearing fictional and made-up.

**Your modifications must include:**

1. **Imaginary Language Design**: - Rename keywords, function names, types, and operators using plausible yet fictional terms. - Preserve the **structure, indentation, and logical flow** of the original code. - Ensure the resulting code remains readable and clearly maps to the original logic.

2. **Consistency and Fidelity**: - Maintain **1-to-1 correspondence** between the original code constructs and their fictional equivalents. - The imaginary language should resemble the **style and design patterns** of the original language (e.g., Pythonic indentation, Java-style braces and semicolons, C++ class structure, etc.).

3. **Creativity within Constraint**: - Make the language feel internally consistent and syntactically plausible. - Avoid random noise—each fictional token should appear intentional and reusable.

**Guidelines:**

- **Do not change the underlying logic** of the original code. - **Do not translate comments or docstrings**—leave them unchanged. - **Do not add explanations, annotations, or headings** outside the code block. - Output only the converted code. - Ensure formatting matches the original exactly (e.g., spacing, newlines).

**Input:** Instruction: {instruction} Original Response: {response}

**Output:**

```imaginary
{{code_in_imaginary_language}}
```

Figure 11: Imaginary language translation prompt. The task is to render real code in a fictional but consistent language without changing its logic.

---

**Rubric Generation Prompt**

You are tasked with generating an instance-specific evaluation rubric based on a given coding prompt, canonical solution, and test case(s) to evaluate the model-generated response.

**Guidelines:**
- The rubric must be **example-specific**: every score level must directly reference the details of the given prompt, canonical solution, and test case(s).
- Use a fixed 1–10 scale (1 = lowest quality attempt, 10 = fully correct).
- Structure the rubric so that:
- Scores 1–3 describe model responses that are irrelevant, nonsensical, or do not implement the required functionality.
- Scores 4–7 describe model responses that attempt the task but are incomplete, flawed, or only partially correct on test case(s).
- Scores 8–10 describe model responses that are mostly or fully correct, aligning with the canonical solution and passing most or all test case(s).
- Each score level (1–10) must have a clear, measurable description unique to this problem.
- Output only the rubric.

**Input:**
Code Prompt:

```
{code_prompt}
```

Canonical Solution:

```
{canonical_solution}
```

Test Case(s):

```
{test_case}
```

**Output:**

```
{{rubric}}
```

Figure 12: LLM-as-judge prompt for generating an instance-specific rubric to evaluate model-generated code responses.

### A.7.3 TASK PERFORMANCE UNDER PERTURBATIONS AGGREGATED BY EXPLICITNESS OF CODE STRUCTURE (RQ2)

**Qwen3 model family results (explicitness of code structure perturbations)**   See performance of aggregated task performance under explicitness of code structure perturbations in Figure 25.

---

**LLM-as-Judge Evaluation Prompt**

You are tasked with evaluating a model-generated response to a coding prompt using the provided rubric.

You are given:
1. The coding prompt.
2. The rubric (instance-specific, with 1–10 levels).
3. The model response.

**Instructions:**
- Carefully read the rubric.
- Compare the model response against the rubric criteria.
- Assign the most appropriate score (1–10).
- Provide a concise justification inside ¡reasoning¿¡/reasoning¿, explicitly referencing how the model response aligns or fails to align with specific rubric levels.
- Provide only the numeric score inside ¡score¿¡/score¿.
- Do not include any text outside the required tags.

**Input:**
Coding Prompt:

```
{code_prompt}
```

Rubric:

```
{rubric}
```

Model Response:

```
{model_response}
```

**Output:**

```
<reasoning>{{concise justification}}</reasoning>
<score>{{integer from 1 to 10}}</score>
```

---

Figure 13: LLM-as-judge prompt for rubric-based evaluation of model-generated code responses.

**Llama-3.2 model family results (explicitness of code structure perturbations)**  See performance of aggregated task performance under explicitness of code structure perturbations in Figure 26.

**Gemma-3 model family results (explicitness of code structure perturbations)**  See performance of aggregated task performance under explicitness of code structure perturbations in Figure 27.

**OlMo-2 model family results (explicitness of code structure perturbations)**  See performance of aggregated task performance under explicitness of code structure perturbations in Figure 28.

**SmolLM2 model family results (explicitness of code structure perturbations)**  See performance of aggregated task performance under explicitness of code structure perturbations in Figure 29.

### A.7.4 TASK PERFORMANCE UNDER PERTURBATIONS AGGREGATED BY RELATIVE INFORMATION DENSITY (RQ2)

**Qwen3 model family results (relative information density perturbations)**  See performance of aggregated task performance under relative information density perturbations in Figure 30.

**Llama-3.2 model family results (relative information density perturbations)**  See performance of aggregated task performance under relative information density perturbations in Figure 31.

**Gemma-3 model family results (relative information density perturbations)**  See performance of aggregated task performance under relative information density perturbations in Figure 32.

**OlMo-2 model family results (relative information density perturbations)**  See performance of aggregated task performance under relative information density perturbations in Figure 33.

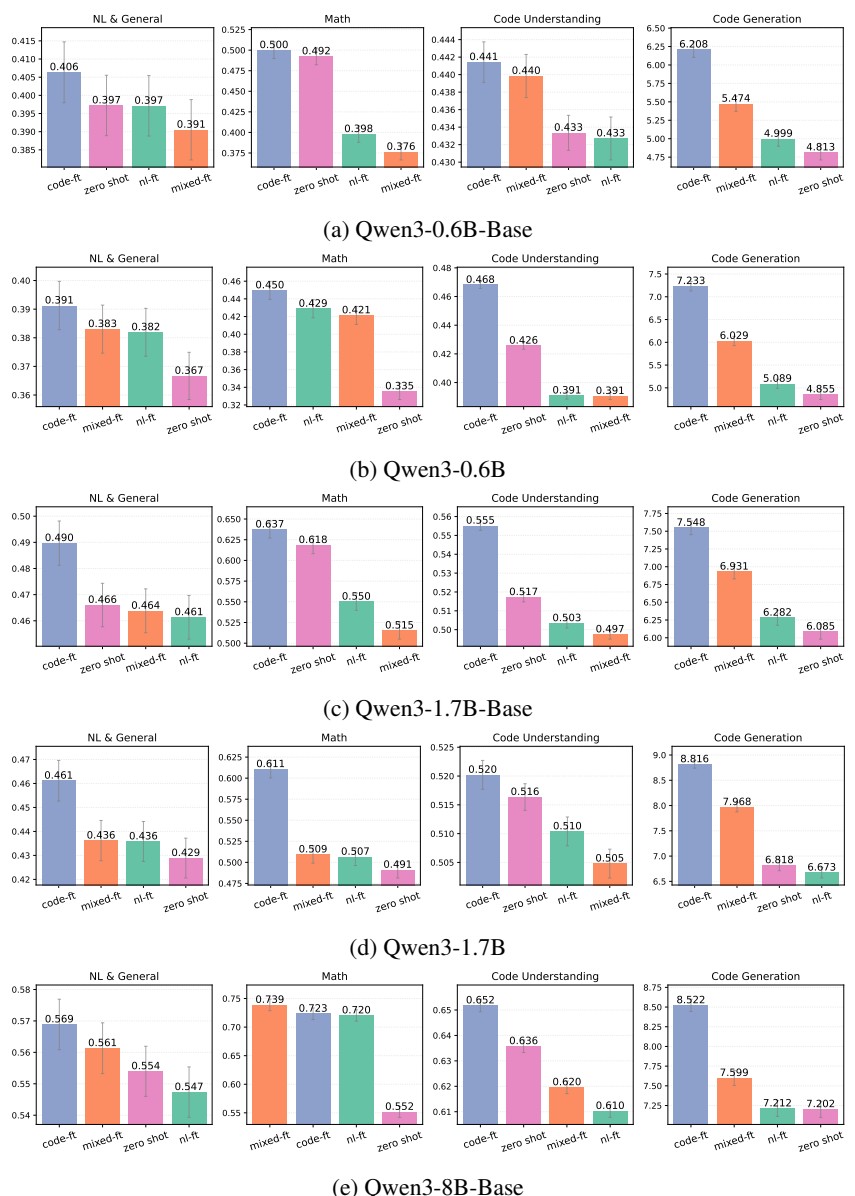

Figure 14: Task performance of Qwen-3 family under zero-shot, full code fine-tuning (code-ft), full natural language fine-tuning (nl-ft), and code-NL mixed fine-tuning (mixed) configurations.

**SmolLM2 model family results (relative information density perturbations)**  See performance of aggregated task performance under relative information density perturbations in Figure 34.

### A.7.5  TASK PERFORMANCE UNDER PERTURBATIONS AGGREGATED BY HUMAN INTERPRETABILITY (RQ2)

**Qwen3 model family results (human interpretability perturbations)**  See performance of aggregated task performance under human interpretability perturbations in Figure 35.

**Llama-3.2 model family results (human interpretability perturbations)**  See performance of aggregated task performance under human interpretability perturbations in Figure 36.

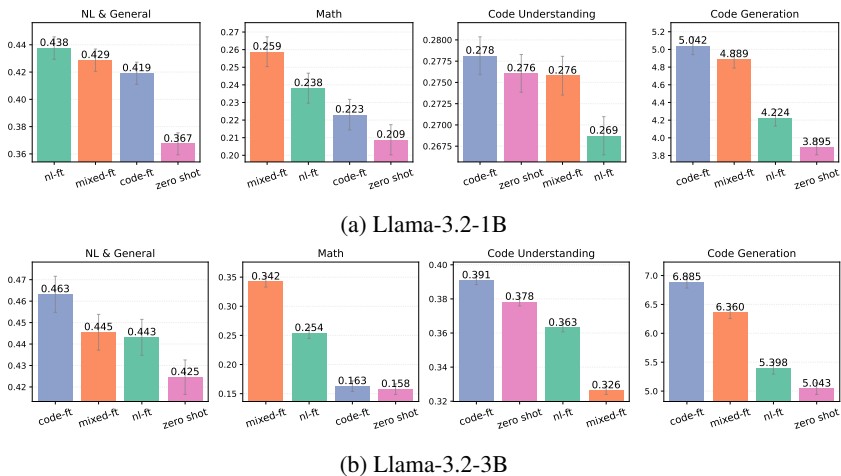

(a) Llama-3.2-1B

(b) Llama-3.2-3B

Figure 15: Task performance of Llama-3.2 family under zero-shot, full code fine-tuning (code-ft), full natural language fine-tuning (nl-ft), and code-NL mixed fine-tuning (mixed) configurations.

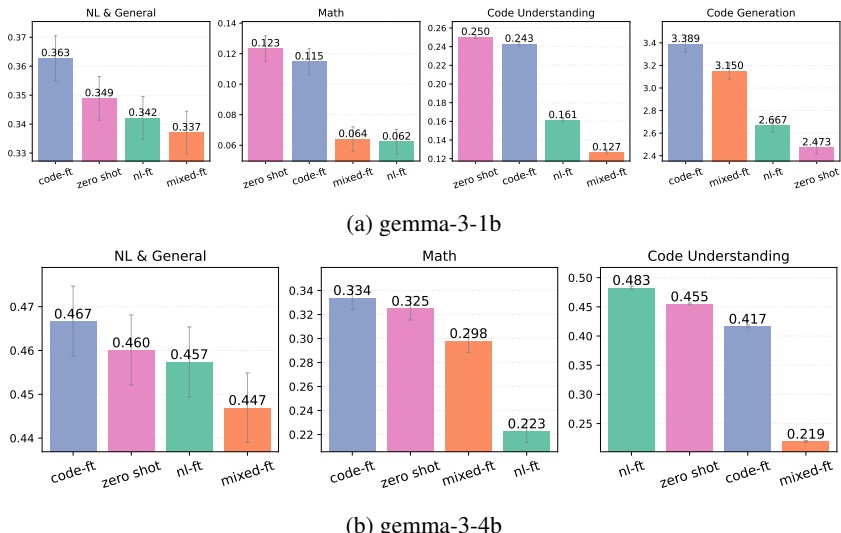

(a) gemma-3-1b

(b) gemma-3-4b

Figure 16: Task performance of Gemma-3 family under zero-shot, full code fine-tuning (code-ft), full natural language fine-tuning (nl-ft), and code-NL mixed fine-tuning (mixed) configurations.

**Gemma-3 model family results (human interpretability perturbations)** See performance of aggregated task performance under human interpretability perturbations in Figure 37.

**OlMo-2 model family results (human interpretability perturbations)** See performance of aggregated task performance under human interpretability perturbations in Figure 38.

**SmolLM2 model family results (human interpretability perturbations)** See performance of aggregated task performance under human interpretability perturbations in Figure 39.

A.7.6  TASK PERFORMANCE FOR ALL INDIVIDUAL PERTURBATIONS (RQ2)

**Qwen3 model family results (individual perturbations)** See performance of all perturbation configurations in Figure 40.

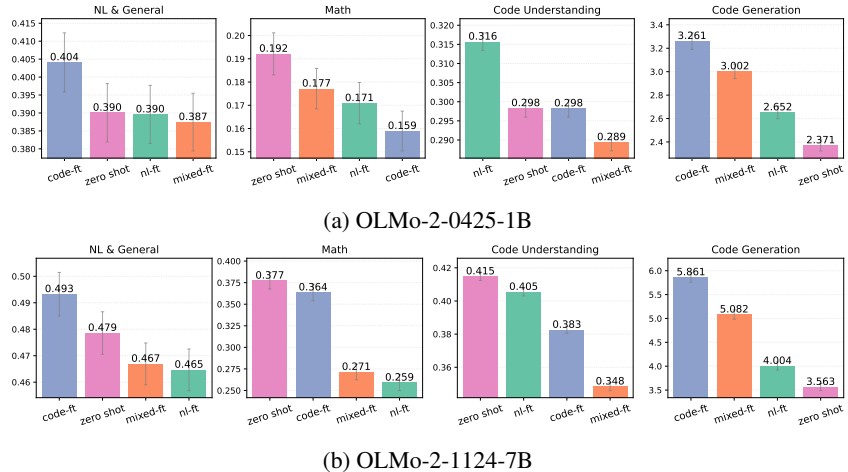

Figure 17: Task performance of OLMo-2 family under zero-shot, full code fine-tuning (code-ft), full natural language fine-tuning (nl-ft), and code-NL mixed fine-tuning (mixed) configurations.

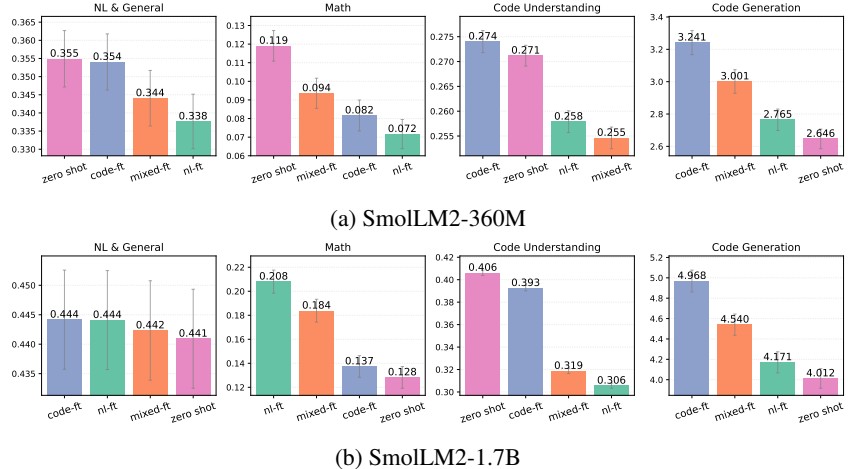

Figure 18: Task performance of SmolLM2 family under zero-shot, full code fine-tuning (code-ft), full natural language fine-tuning (nl-ft), and code-NL mixed fine-tuning (mixed) configurations.

**Llama-3.2 model family results (individual perturbations)** See performance of all perturbation configurations in Figure 41.

**Gemma-3 model family results (individual perturbations)** See performance of all perturbation configurations in Figure 42.

**OlMo-2 model family results (individual perturbations)** See performance of all perturbation configurations in Figure 43.

**SmolLM2 model family results (individual perturbations)** See performance of all perturbation configurations in Figure 44.

A.7.7   TASK PERFORMANCE WITH DIFFERENT PROGRAMMING LANGUAGES (RQ3)

**Qwen3 model family results** See performance of grouped performance and individual programming languages in Figure 45 and Figure 46, respectively.

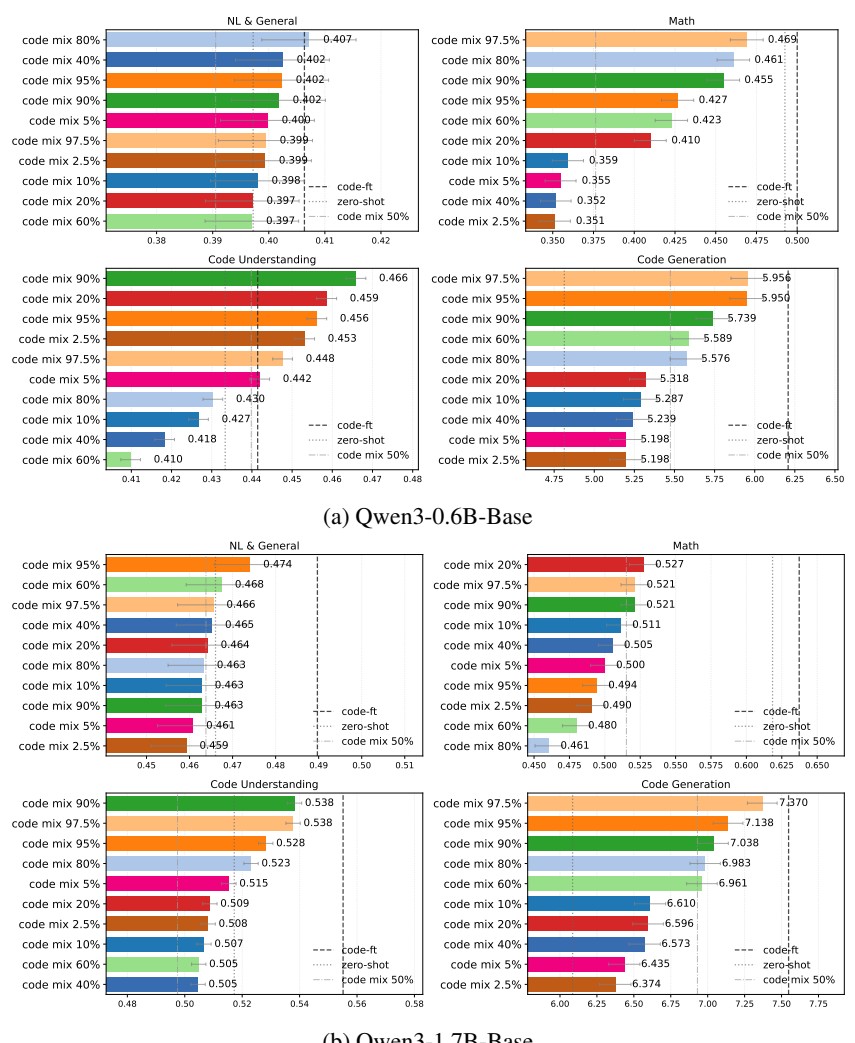

(a) Qwen3-0.6B-Base

(b) Qwen3-1.7B-Base

Figure 19: Task performance of Qwen3-0.6, 1.7B-Base when mixing different ratio of code data during fine-tuning . In general higher code percentages improves performance, with math tasks showing large variation.

**Llama-3 model family results** See performance of grouped performance and individual programming languages in Figure 47.

**SmolLM2 model family results** See performance of grouped performance and individual programming languages in Figure 48.

### A.7.8 LLM-AS-JUDGE RESULTS

We report the results across multiple judge models in Table 7.

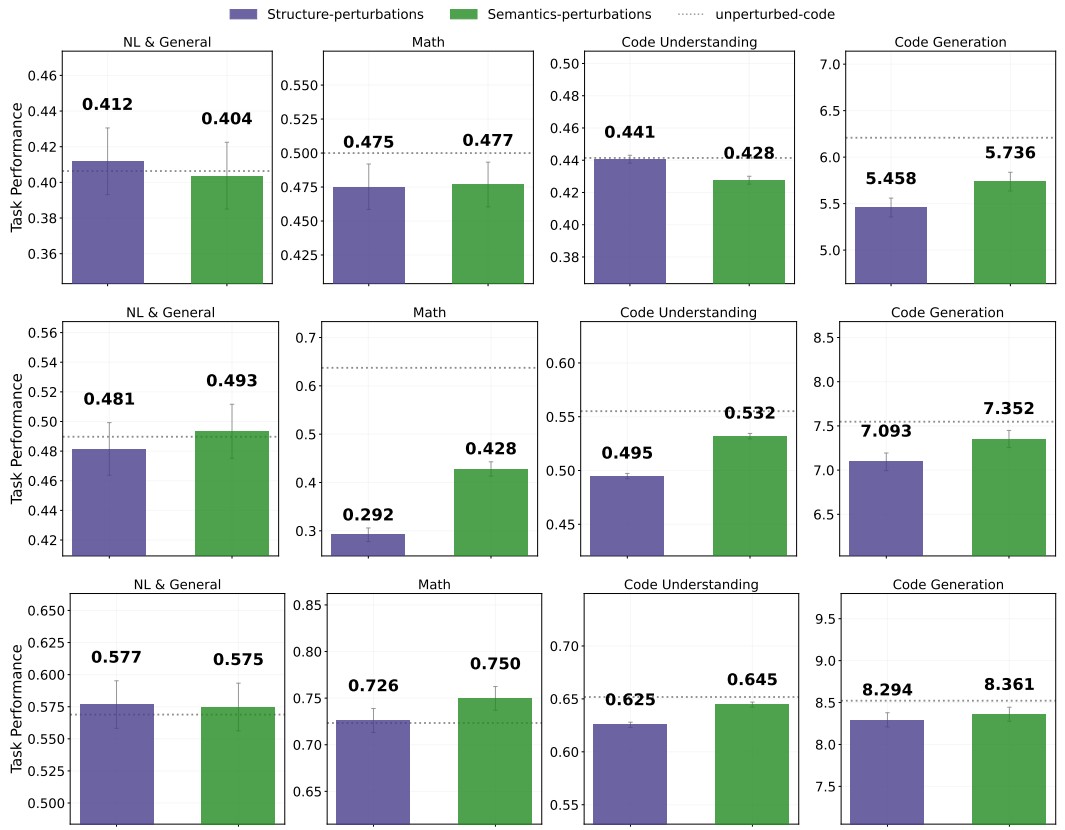

Figure 20: Task performance under perturbations aggregated by structure vs semantics across Qwen3-Base models (0.6B (top), 1.7B (mid), 8B (bottom)).

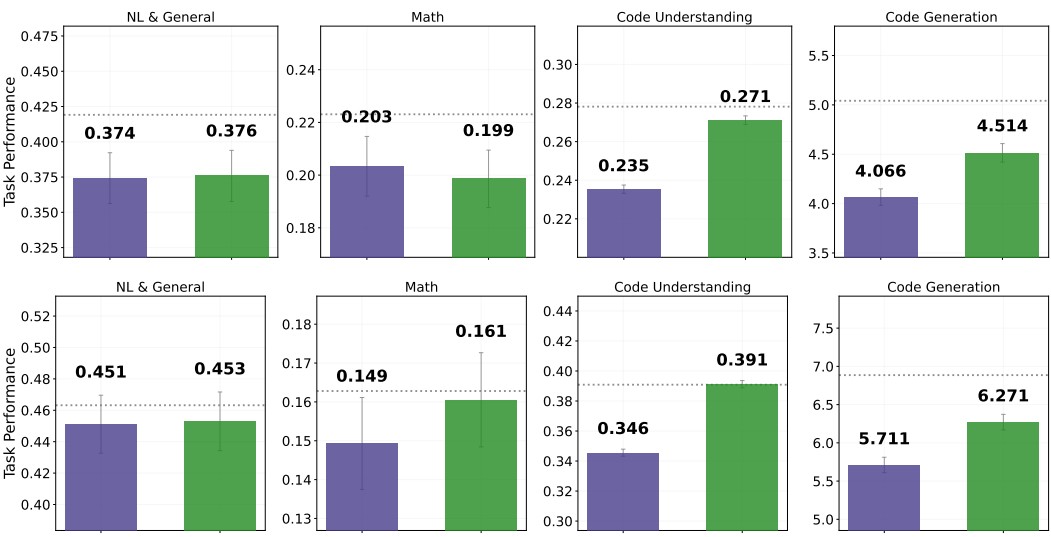

Figure 21: Task performance under perturbations aggregated by structure vs semantics across Llama-3.2 models (1B (top), 3B (bottom)).

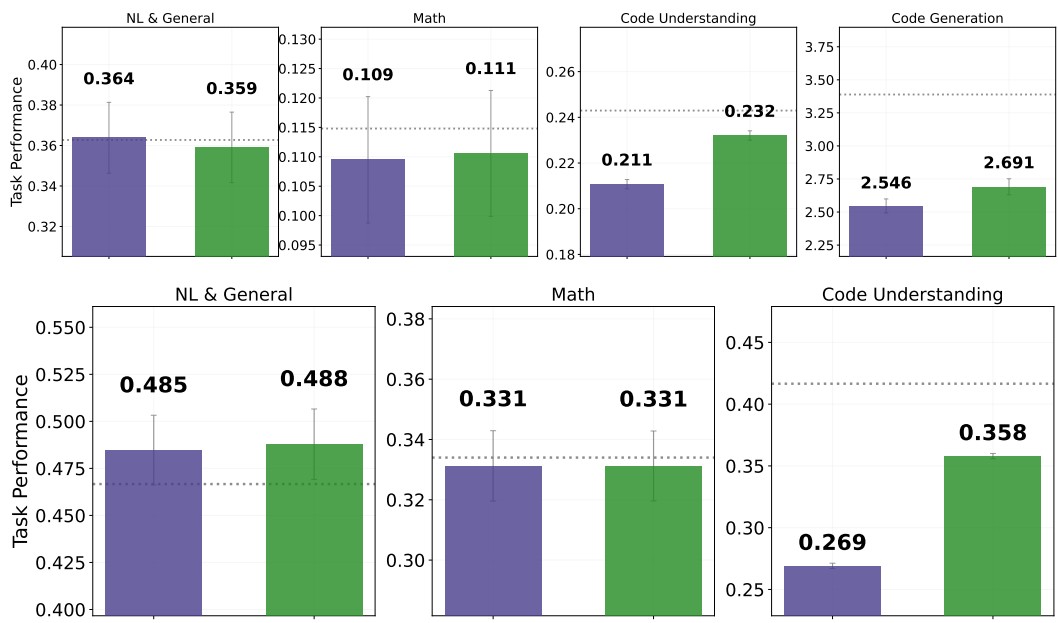

Figure 22: Task performance under perturbations aggregated by structure vs semantics across Gemma-3 models (1B (top), 4B (bottom)).

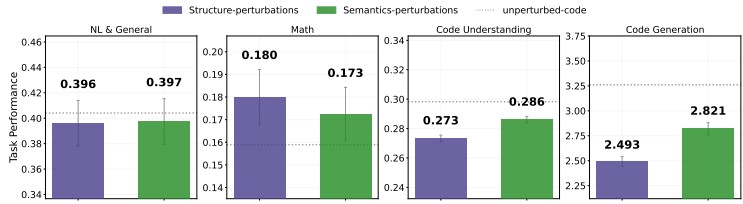

Figure 23: Additional performance of OLMo-2-0425-1B aggregated by structure vs semantics across tasks.

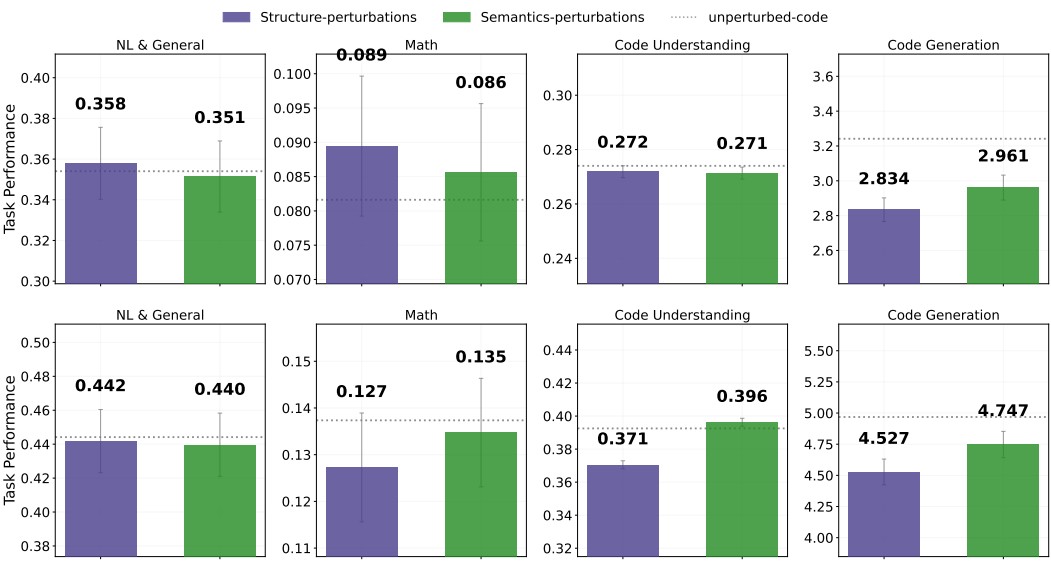

Figure 24: Task performance under perturbations aggregated by structure vs semantics across SmolLM2 models (360M (top), 1.7B (bottom)).

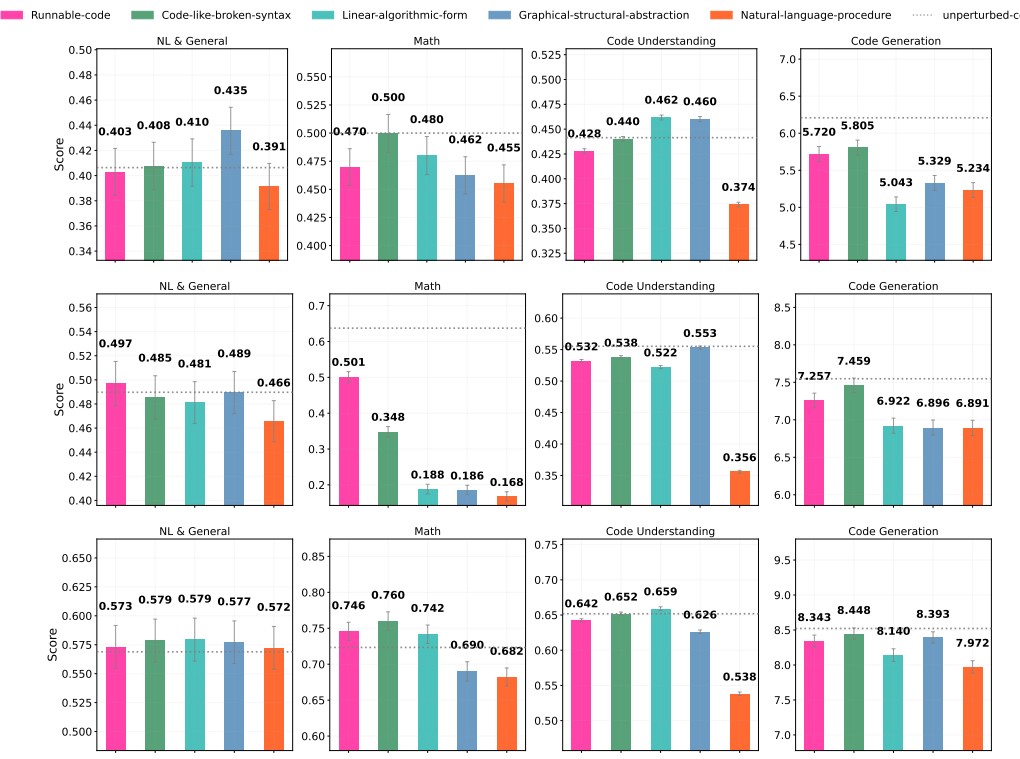

Figure 25: Task performance under perturbations aggregated by explicitness of code structure across Qwen3-Base models (0.6B (top), 1.7B (mid), 8B (bottom)).

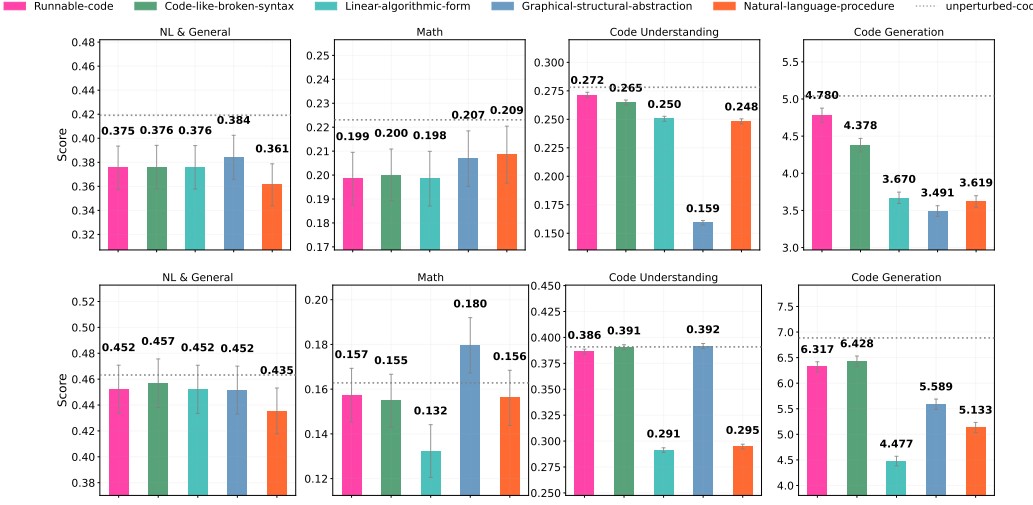

Figure 26: Task performance under perturbations aggregated by explicitness of code structure across Llama-3.2 models (1B (top), 3B (bottom)).

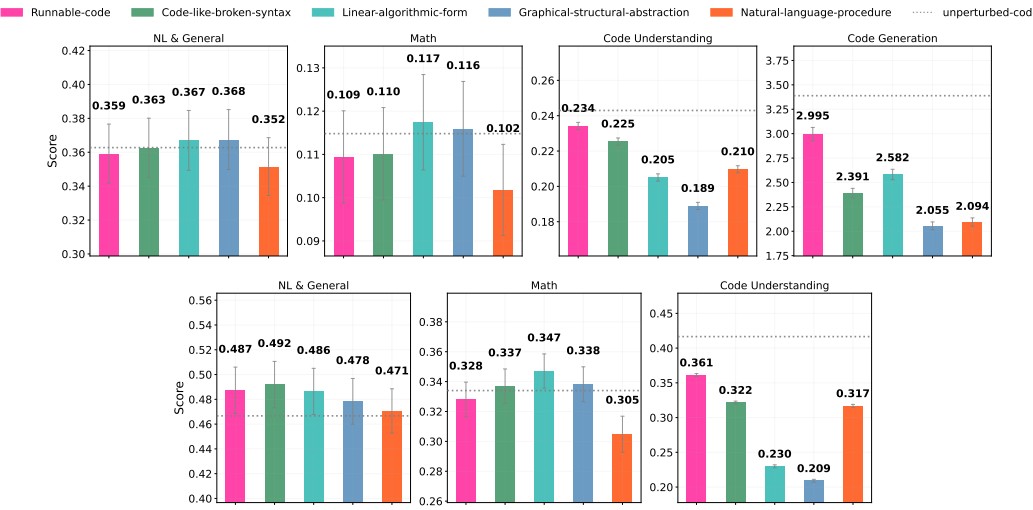

Figure 27: Task performance under perturbations aggregated by explicitness of code structure across Gemma-3 models (1B (top), 4B (bottom)).

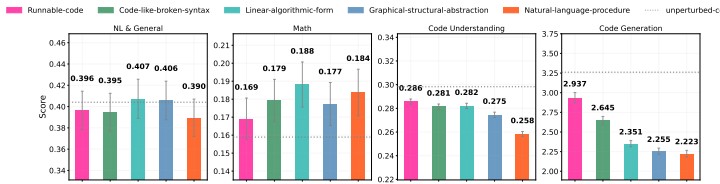

Figure 28: Additional performance of OLMo-2-0425-1B aggregated by explicitness of code structure across tasks.

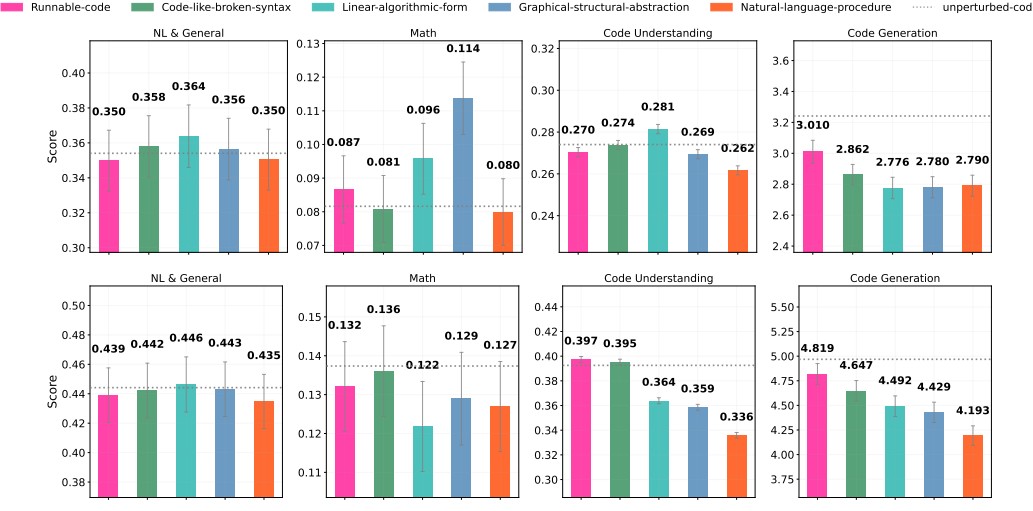

Figure 29: Task performance under perturbations aggregated by explicitness of code structure across SmolLM2 models (360M (top), 1.7B (bottom)).

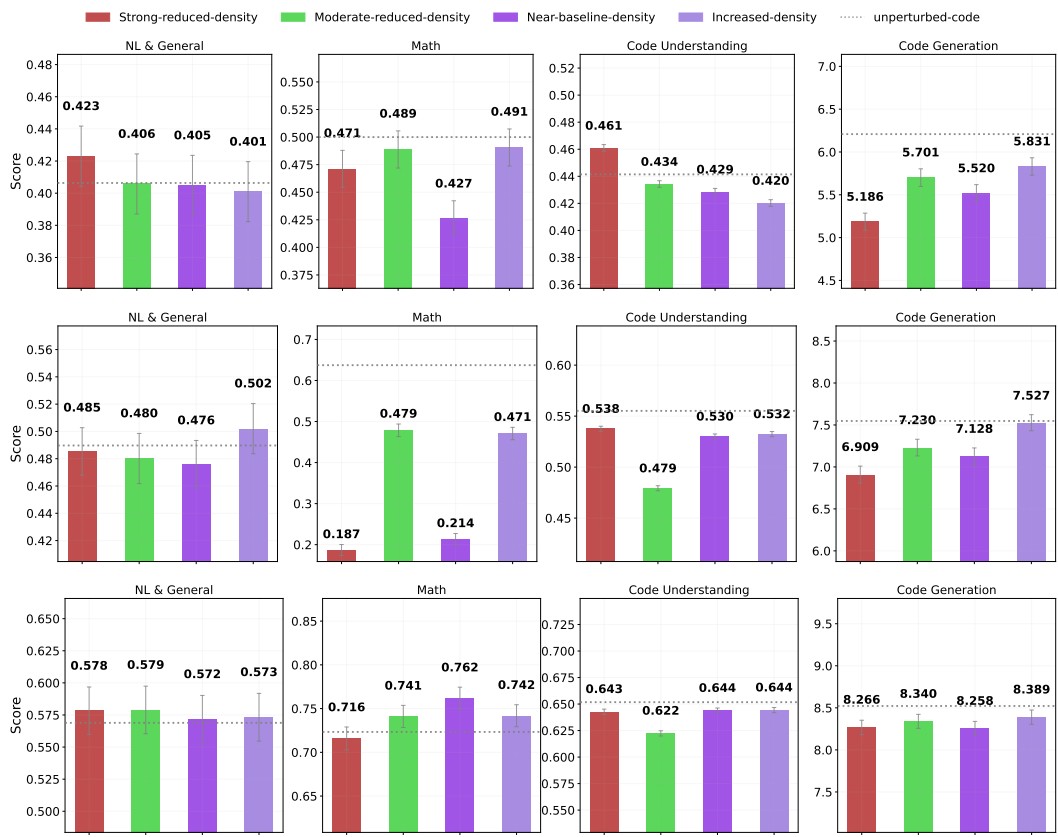

Figure 30: Task performance under perturbations aggregated by relative information density across Qwen3-Base models (0.6B (top), 1.7B (mid), 8B (bottom)).

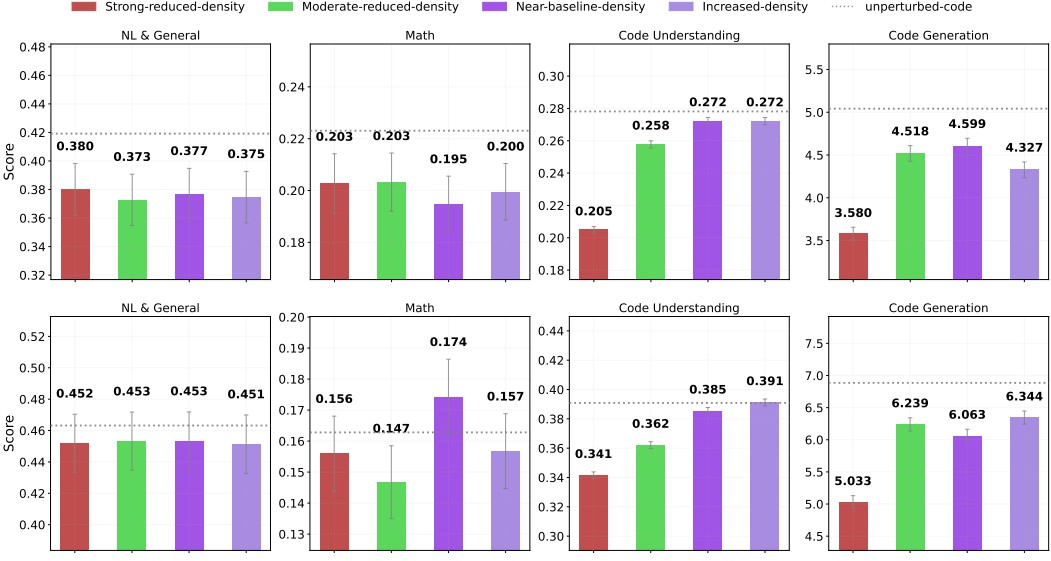

Figure 31: Task performance under perturbations aggregated by relative information density across Llama-3.2 models (1B (top), 3B (bottom)).

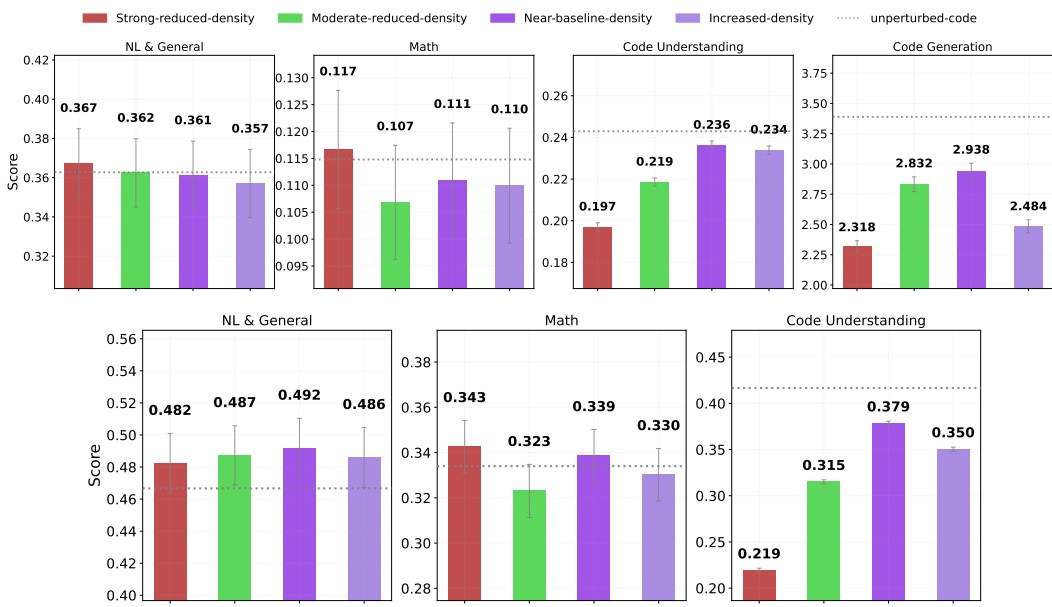

Figure 32: Task performance under perturbations aggregated by relative information density across Gemma-3 models (1B (top), 4B (bottom)).

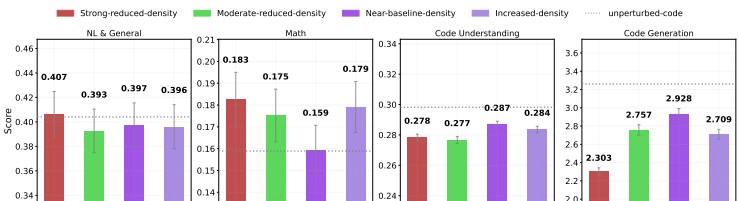

Figure 33: Additional performance of OLMo-2-0425-1B aggregated by relative information density across tasks.

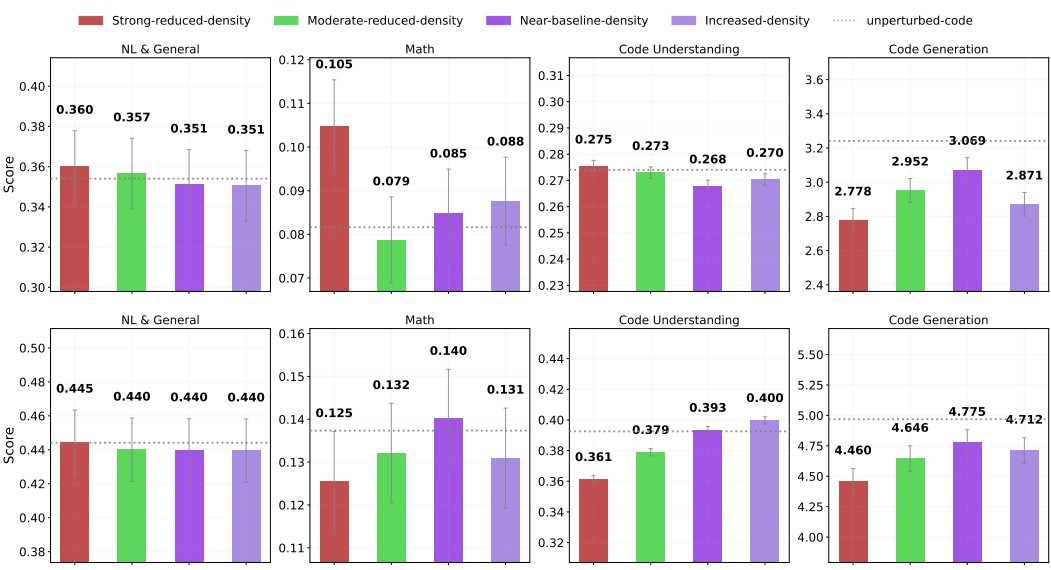

Figure 34: Task performance under perturbations aggregated by relative information density across SmolLM2 models (360M (top), 1.7B (bottom)).

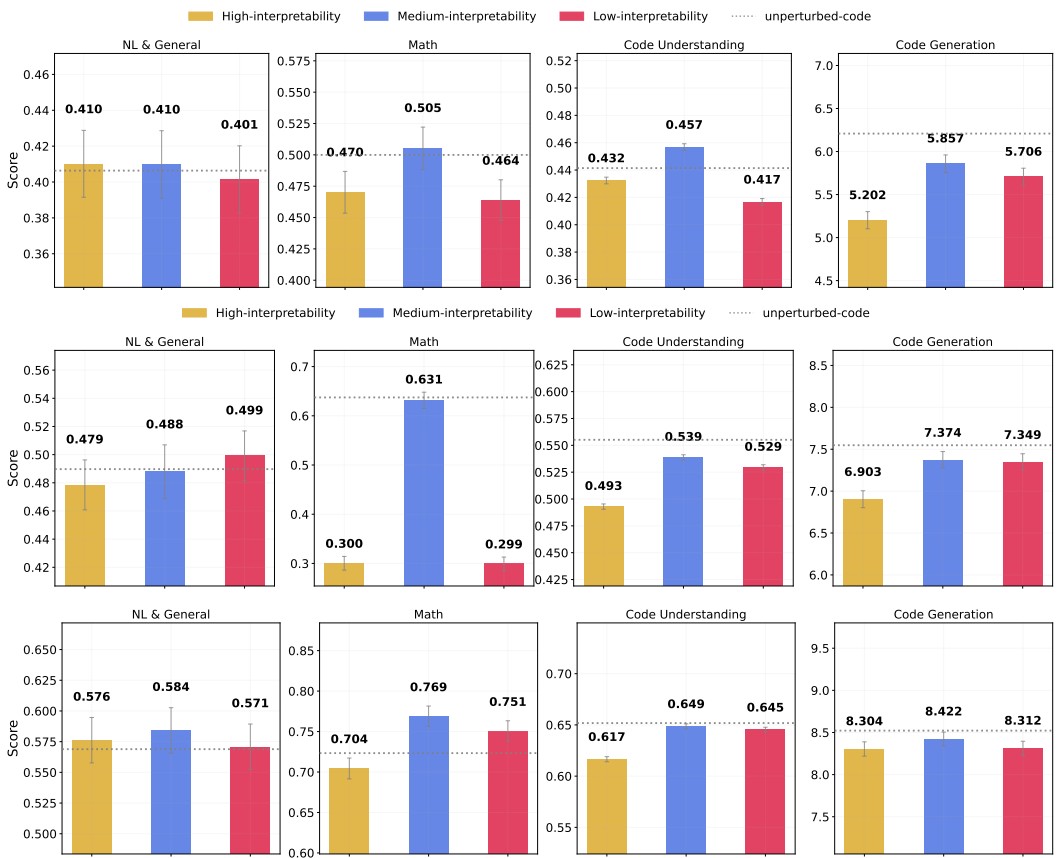

Figure 35: Task performance under perturbations aggregated by human interpretability across Qwen3-Base models (0.6B (top), 1.7B (mid), 8B (bottom)).

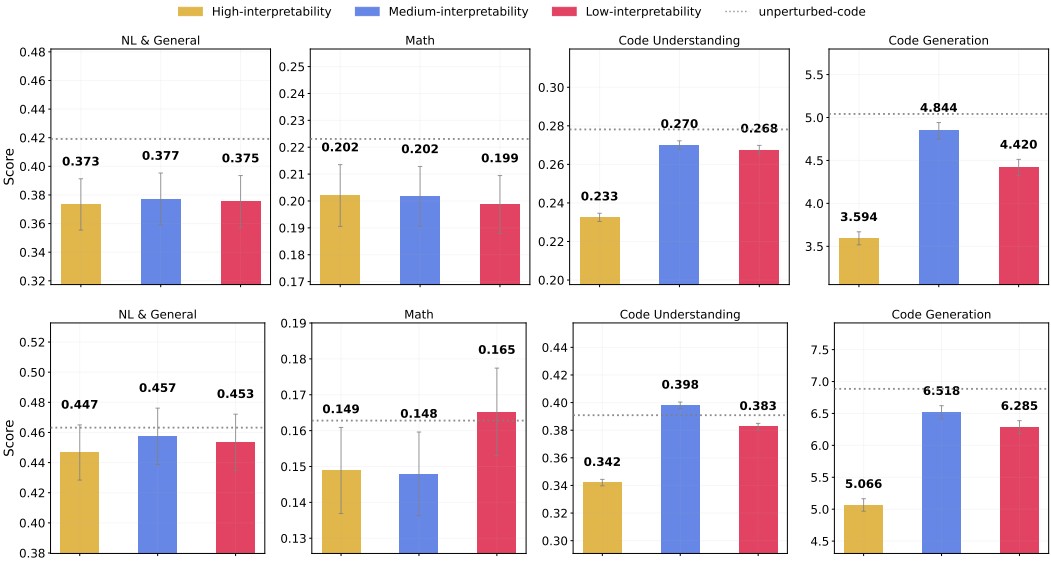

Figure 36: Task performance under perturbations aggregated by human interpretability across Llama-3.2 models (1B (top), 3B (bottom)).

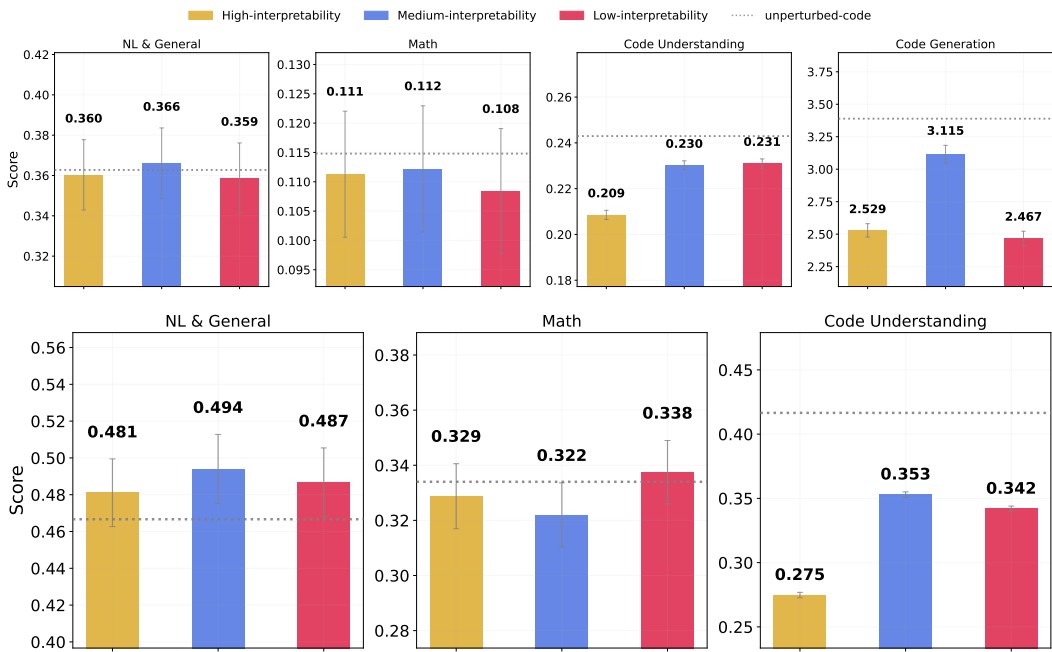

Figure 37: Task performance under perturbations aggregated by human interpretability across Gemma-3 models (1B (top), 4B (bottom)).

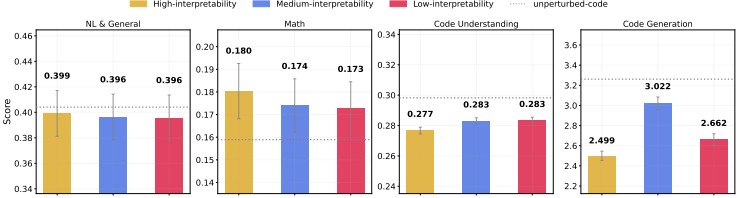

Figure 38: Additional performance of OLMo-2-0425-1B aggregated by human interpretability across tasks.

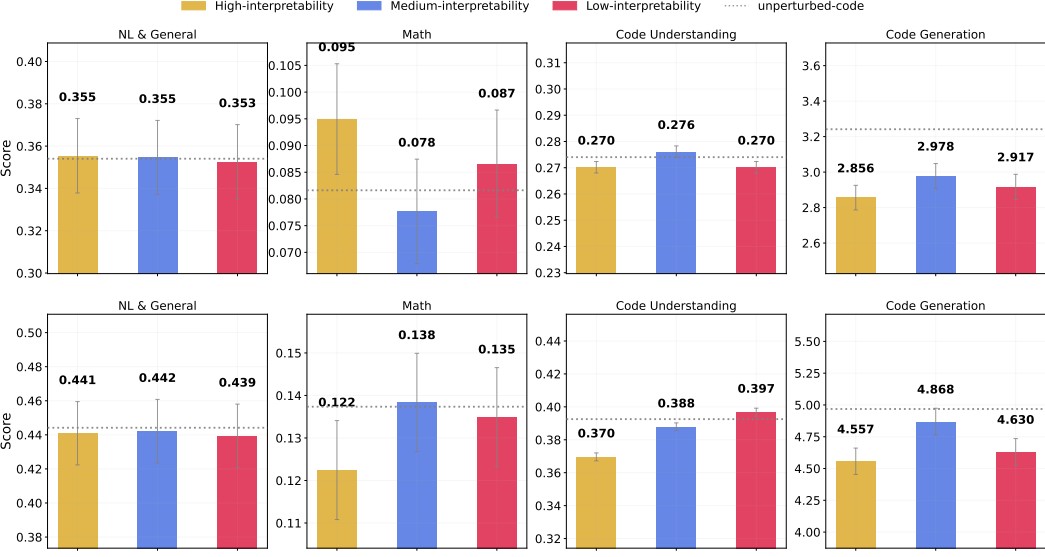

Figure 39: Task performance under perturbations aggregated by human interpretability across SmolLM2 models (360M (top), 1.7B (bottom)).

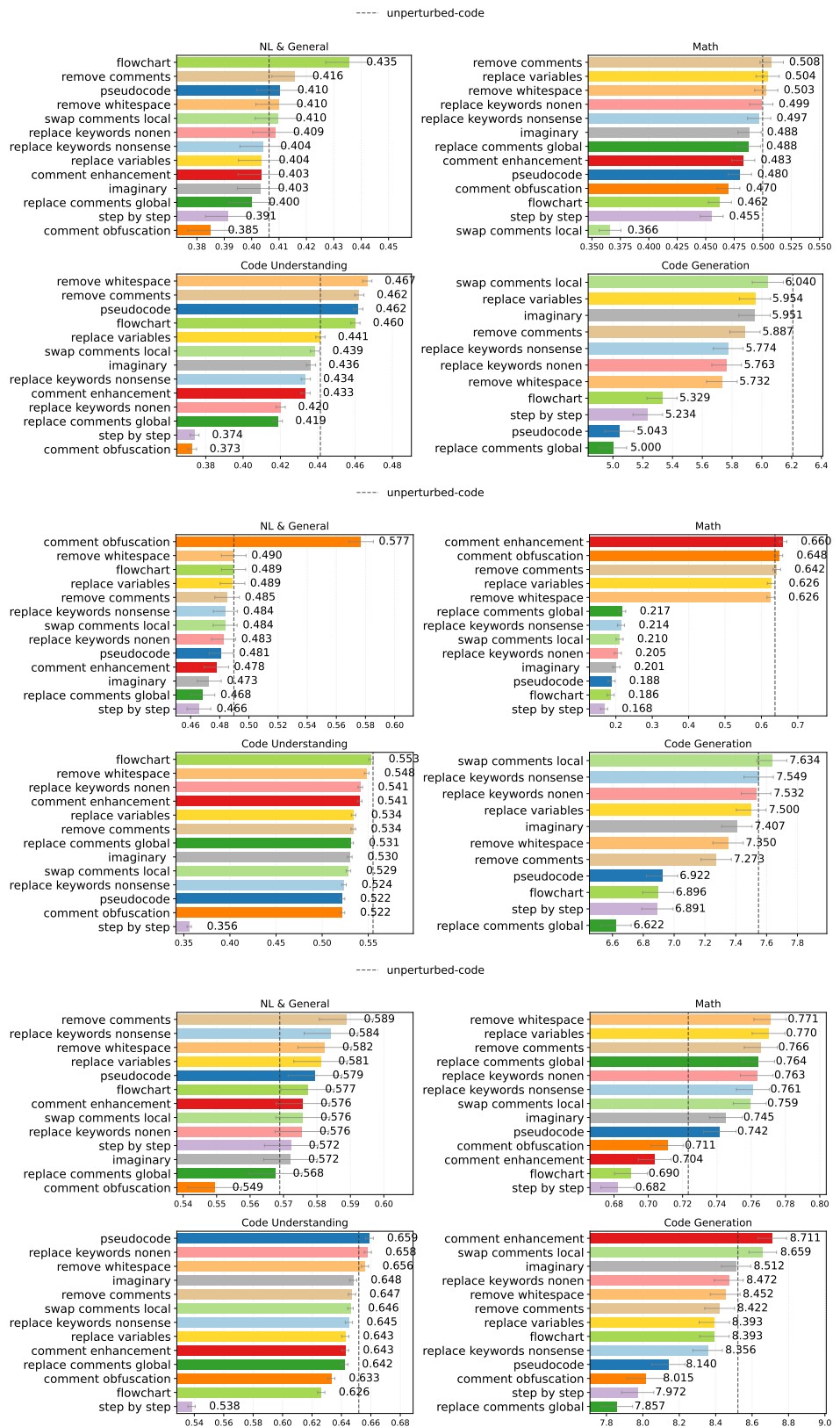

Figure 40: All perturbations across Qwen3-Base models (0.6B (top), 1.7B (mid), 8B (bottom)).

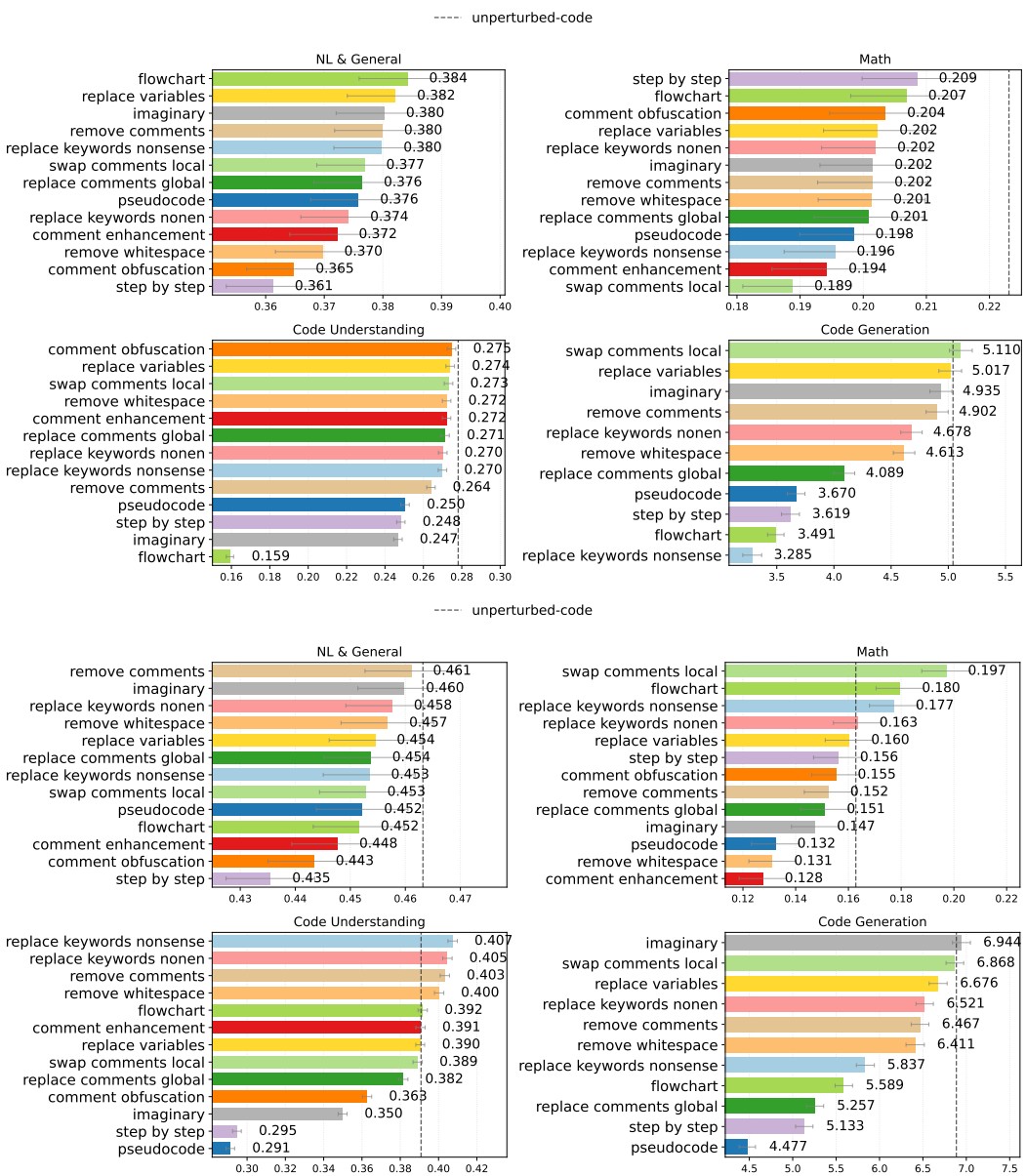

Figure 41: All perturbations across Llama-3.2 models (1B (top), 3B (bottom)).

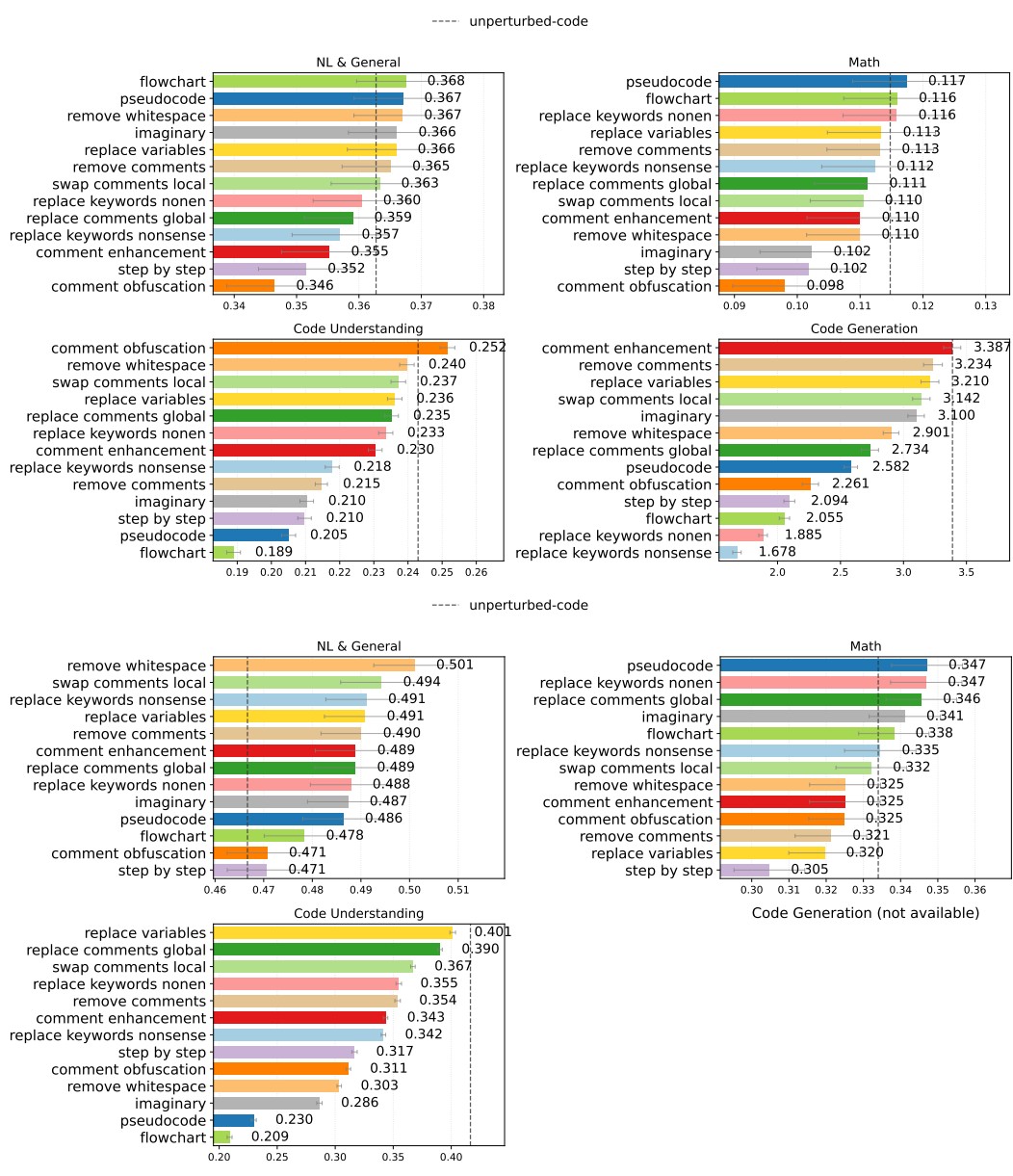

Figure 42: All perturbations across Gemma-3 models (1B (top), 4B (bottom)).

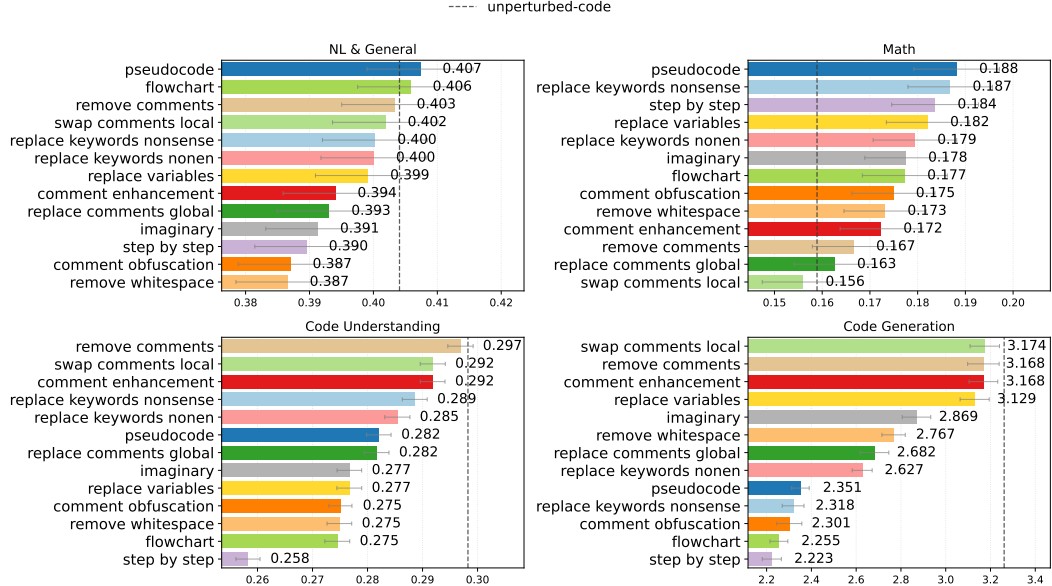

Figure 43: OLMo-2-0425-1B with all perturbations.

Table 7: Cross-judge evaluation of Qwen3-4B base variant on Python code generation task using five LLM judges — all under identical evaluation settings. Model rankings remain consistent across judges, with only moderate score variability (std 0.63–0.99), demonstrating that LLM-as-judge evaluations are stable and reliable across different judging models.

| Target Model / Perturbation | claude-3-haiku | claude-haiku-4.5 | gpt-4o-mini | gpt-5-mini | llama3-90b | Mean | Std |
|---|---|---|---|---|---|---|---|
| zero-shot | 8.41 ± 2.42 | 7.34 ± 2.99 | 7.09 ± 3.27 | 6.84 ± 3.00 | 8.17 ± 2.59 | 7.57 | 0.69 |
| swap_comments_global | 9.01 ± 1.41 | 7.76 ± 2.43 | 7.91 ± 2.55 | 6.85 ± 2.94 | 9.13 ± 1.42 | 8.13 | 0.95 |
| swap_comments_local | 9.24 ± 1.24 | 7.94 ± 2.69 | 8.54 ± 2.37 | 7.26 ± 3.06 | 9.15 ± 1.58 | 8.43 | 0.84 |
| replace_keywords_nonsense | 9.10 ± 1.16 | 7.88 ± 2.46 | 8.74 ± 2.31 | 7.16 ± 3.04 | 9.20 ± 1.39 | 8.42 | 0.87 |
| replace_keywords_nonEn | 9.21 ± 1.16 | 7.68 ± 2.70 | 8.73 ± 2.22 | 7.35 ± 2.89 | 9.25 ± 1.20 | 8.44 | 0.88 |
| flowchart | 8.88 ± 1.67 | 7.44 ± 2.86 | 8.07 ± 2.58 | 7.32 ± 3.01 | 9.13 ± 1.50 | 8.17 | 0.82 |
| imaginary | 8.94 ± 1.68 | 7.67 ± 2.87 | 8.06 ± 2.71 | 7.17 ± 2.94 | 9.00 ± 1.82 | 8.17 | 0.80 |
| pseudocode | 8.89 ± 1.62 | 7.15 ± 2.72 | 7.34 ± 3.00 | 7.22 ± 2.90 | 9.02 ± 1.48 | 7.92 | 0.94 |
| step_by_step | 8.66 ± 1.92 | 7.50 ± 2.83 | 7.76 ± 2.75 | 7.57 ± 2.90 | 8.81 ± 1.90 | 8.06 | 0.63 |
| comment_obfuscation | 8.79 ± 2.03 | 7.41 ± 2.80 | 7.86 ± 2.88 | 7.02 ± 3.07 | 8.93 ± 1.78 | 8.00 | 0.84 |
| comment_enhancement | 9.22 ± 1.35 | 7.77 ± 2.74 | 8.36 ± 2.65 | 7.93 ± 2.48 | 9.09 ± 1.71 | 8.47 | 0.66 |
| remove_comments | 9.09 ± 1.61 | 7.86 ± 2.54 | 8.27 ± 2.54 | 7.23 ± 3.05 | 9.19 ± 1.42 | 8.32 | 0.83 |
| remove_whitespace | 9.10 ± 1.43 | 7.87 ± 2.58 | 8.69 ± 2.28 | 7.77 ± 2.85 | 9.33 ± 1.33 | 8.55 | 0.71 |
| replace_variables | 9.20 ± 0.95 | 7.47 ± 2.74 | 8.09 ± 2.66 | 6.92 ± 3.09 | 9.07 ± 1.53 | 8.15 | 0.99 |
| code-ft | 9.14 ± 1.46 | 7.98 ± 2.32 | 8.65 ± 2.36 | 7.61 ± 2.98 | 9.17 ± 1.41 | 8.51 | 0.70 |
| mixed-ft | 8.68 ± 1.94 | 7.46 ± 2.94 | 7.94 ± 2.79 | 7.31 ± 3.08 | 8.75 ± 2.10 | 8.03 | 0.67 |
| nl-ft | 8.19 ± 2.43 | 6.58 ± 2.99 | 6.75 ± 3.11 | 6.60 ± 3.15 | 7.84 ± 2.63 | 7.19 | 0.76 |

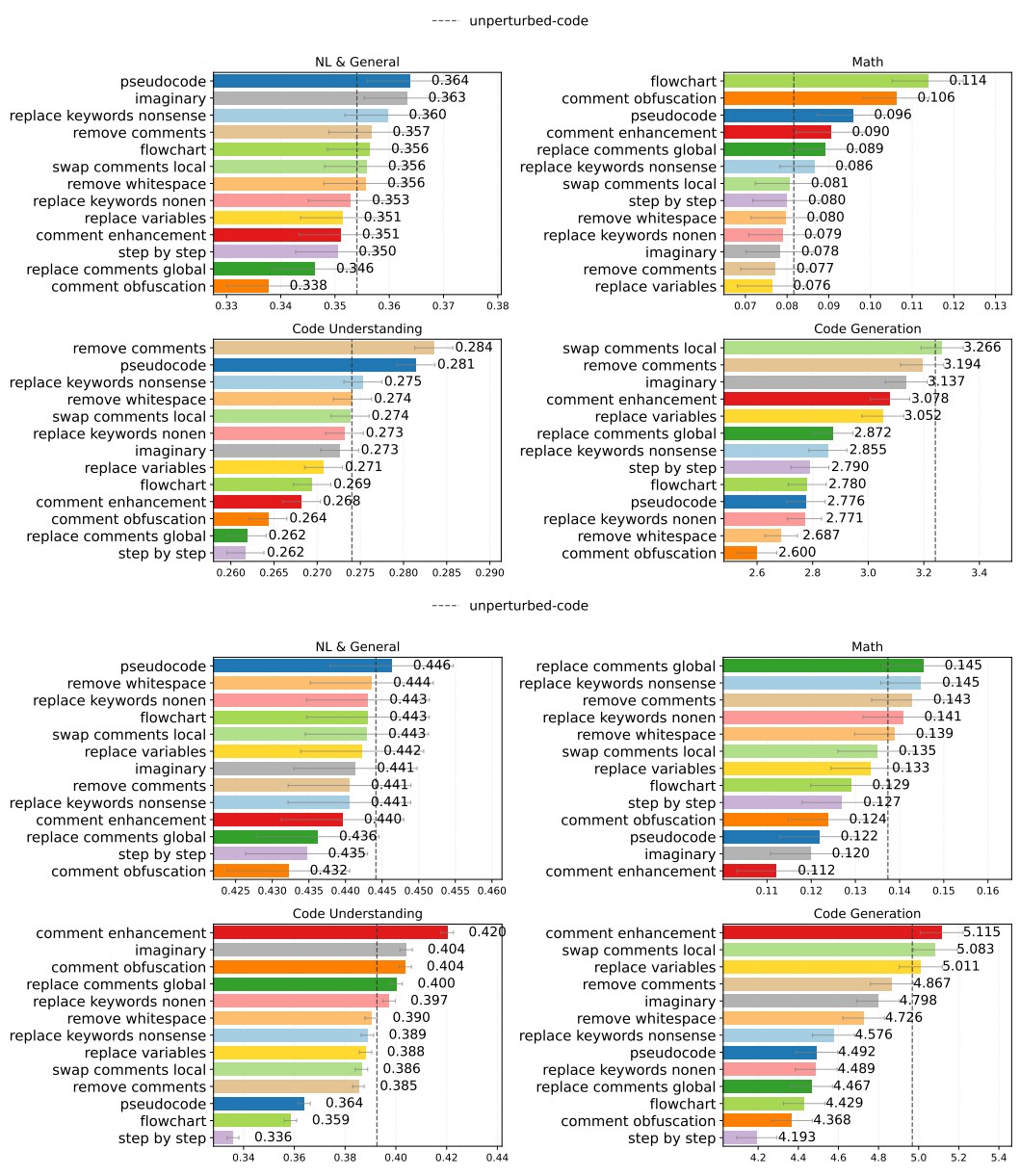

Figure 44: All perturbations across SmolLM2 models (360M (top), 1.7B (bottom)).

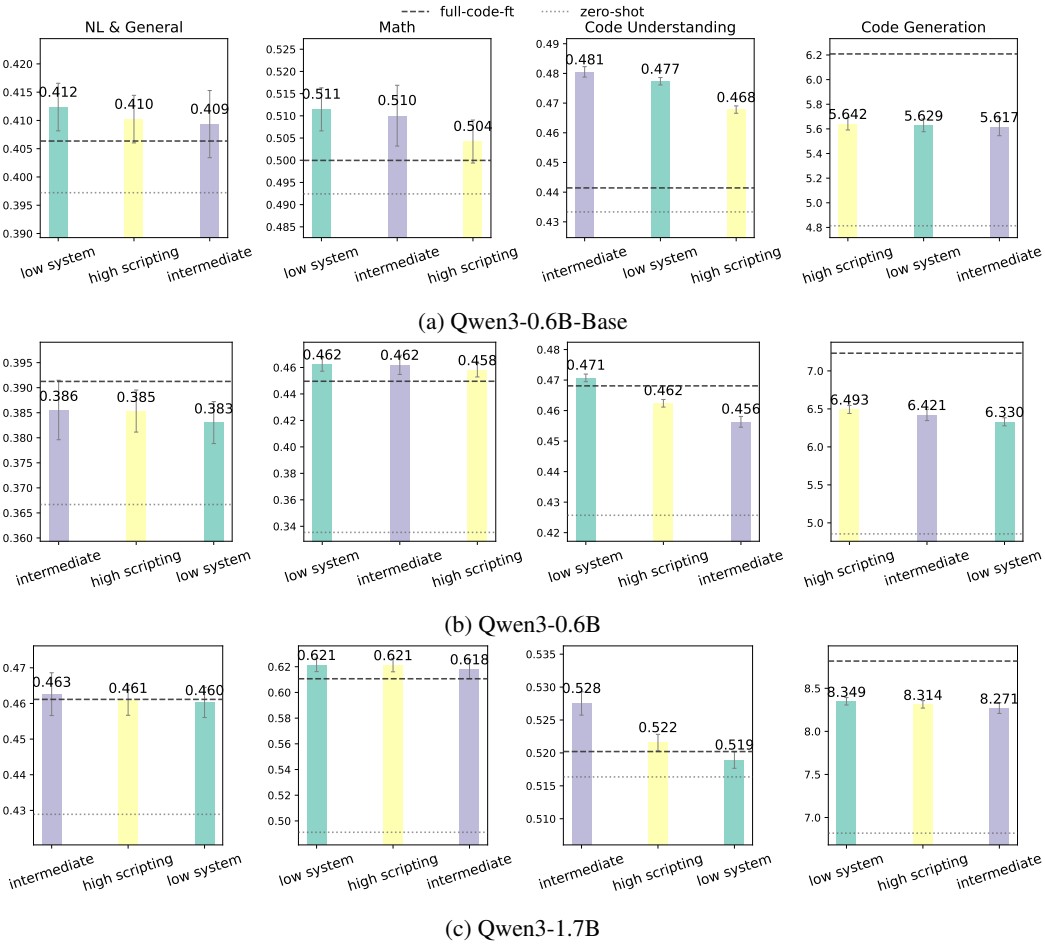

Figure 45: Grouped performance of Qwen-3 family under low-system, intermediate, and high-scripting programming languages.

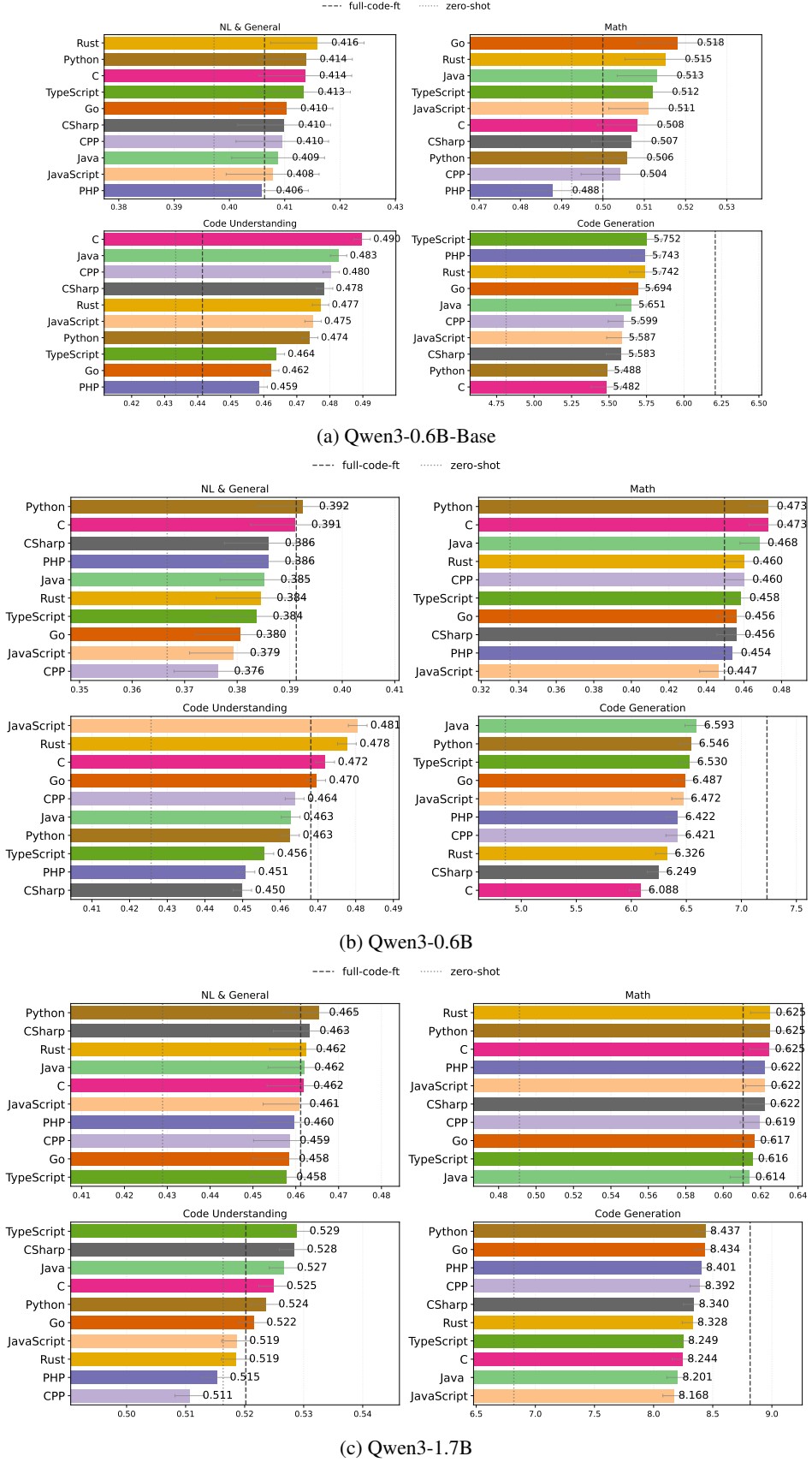

Figure 46: All programming language specific performance of Qwen-3 family.

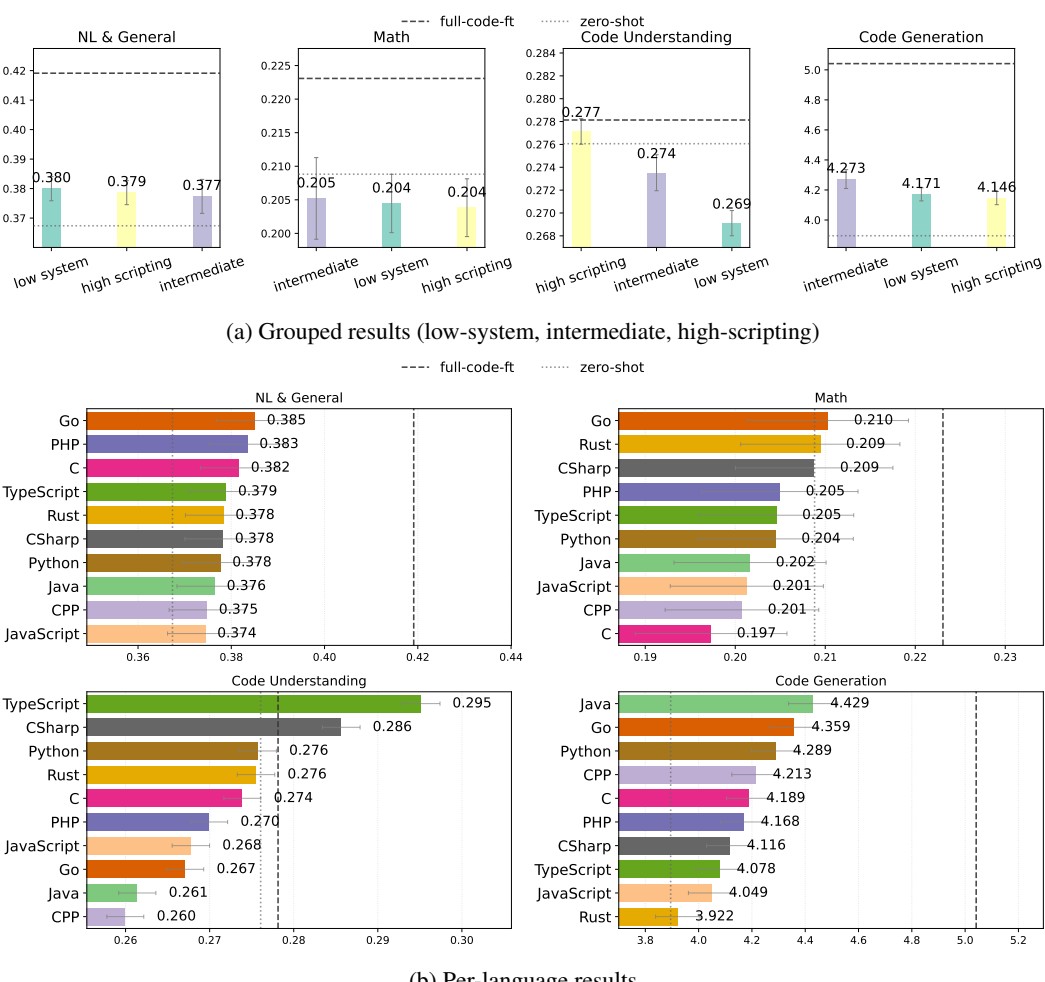

(a) Grouped results (low-system, intermediate, high-scripting)

(b) Per-language results

Figure 47: Performance for Llama-3.2-1B. (a) Programming language groups, (b) individual languages.

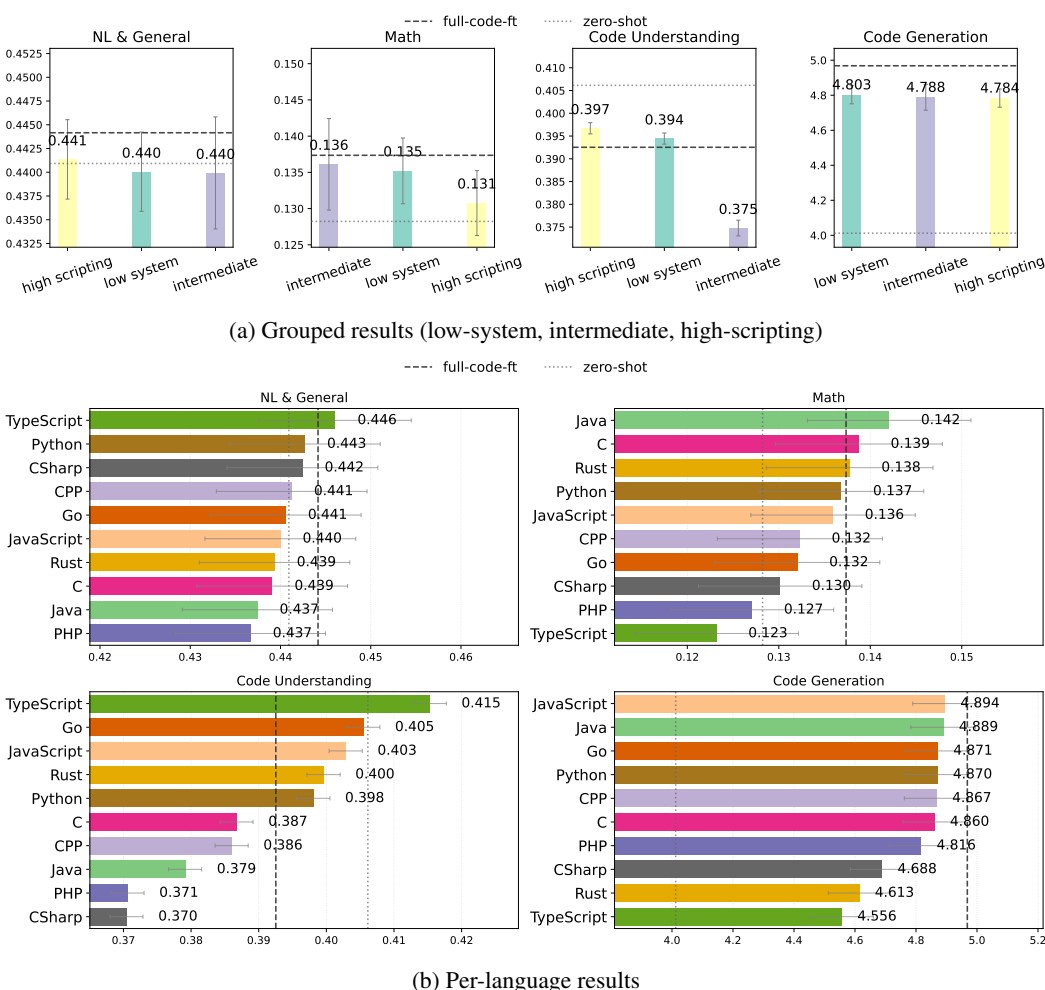

(a) Grouped results (low-system, intermediate, high-scripting)

(b) Per-language results

Figure 48: Performance for SmolLM2-1.7B. (a) Programming language groups, (b) individual languages.

