# OpenReview forum: "On Code-Induced Reasoning in LLMs"
_ICLR.cc/2026/Conference — ICLR 2026 Poster_

### Official Review · Reviewer_DDwQ · 2025-11-01

**Soundness:** 2
**Presentation:** 3
**Contribution:** 2
**Rating:** 4
**Confidence:** 4

**Summary:**

This paper presents a systematic investigation into how different properties of code data influence the reasoning capabilities of large language models (LLMs) during fine-tuning. The authors construct parallel instruction datasets in natural language and across ten programming languages, then apply controlled perturbations targeting structural (e.g., whitespace removal, pseudocode conversion) and semantic (e.g., variable renaming, comment modifications) properties of code. Through lots of fine-tuning experiments spanning varied model families at different scales, they evaluate performance on natural language, math, and code tasks. Key findings reveal that structural perturbations degrade performance more severely than semantic ones, particularly on math and code tasks, while appropriate abstractions like pseudocode and flowcharts can match or exceed the effectiveness of original source code.

**Strengths:**

+ The paper presents an interesting study to understand the impact of code data for LLM reasoning
+ The experiments are conducted in a comprehensive way that include multiple different models with varied settings

**Weaknesses:**

**Flawed Experimental Design - Post-Training Phase Limits Causal Claims About Code Impact on Reasoning**

A fundamental limitation of this work is that all experiments are conducted during the supervised fine-tuning phase on base models already extensively pretrained on massive corpora containing both natural language and code. This design weaknes the paper's central claims about which code properties enhance reasoning, as the observed performance differences may primarily reflect how well different formats activate existing pretrained knowledge rather than demonstrating the intrinsic learning value of code properties. For instance, when fine-tuning on perturbed code (e.g., whitespace removed, variables renamed), poor performance could simply indicate distributional mismatch with pretraining data rather than proving that structural properties are fundamentally important for reasoning. The surprising finding that "corrupted code with misleading signals remains competitive" may merely show that fine-tuning cannot override deeply ingrained pretraining patterns rather than demonstrating genuine robustness to surface regularities. This limitation calls into question whether the paper's guidance for "design of training data" will generalize broadly.

**Correctness in the code data is not guaranteed, and the structural perturbations are overly simplified**

Another weakness of this work is the lack of validation for code correctness in the training data and the oversimplified nature of perturbations, which significantly limits the validity and depth of the findings. The authors generate all code responses using GPT-4o-mini, a relatively weak model, and seems apply no execution-based validation to ensure correctness (or I missed this part?). Without test cases to reject incorrect solutions during data generation, the 120K training examples may contain substantial amounts of buggy or non-functional code, fundamentally compromising the study's premise. If models learn from a majority of flawed examples, observed performance patterns cannot reliably indicate which code properties enhance reasoning but rather which properties are less affected by learning from errors.

Moreover, the absence of test cases severely constrains the types of perturbations possible, forcing reliance on superficial transformations like whitespace removal and variable renaming that preserve surface syntax but cannot guarantee semantic equivalence. The study cannot perform meaningful semantically-equivalent transformations such as algorithmic refactoring, loop-to-recursion conversion, or function extraction that would provide richer insights into whether models learn genuine computational logic versus syntactic templates.

**Inadequate Analysis of Why Pure Code Fine-tuning Outperforms Mixed Training in RQ1**

The paper lacks an in-depth analysis of why pure code fine-tuning (code-ft) consistently outperforms or matches mixed training (mixed-ft) across most tasks in RQ1, despite mixed training intuitively seeming more aligned with evaluation on diverse task types. Several plausible hypotheses remain unexplored, including task distribution mismatch, data quality difference among data resources, or the observations are mostly for post-training? Without ablation studies and analysis to distinguish between these possibilities, readers cannot assess whether the findings reflect fundamental properties of code for reasoning or other factors.

**Questions:**

- Can the authors provide analysis or at least some conceptual arguments to disentangle pretraining effects from intrinsic code properties? Or the authors would discuss the findings specifically for post-training rather than general LLM training (including all training phases)?

- What steps were taken to ensure code correctness in the training data to avoid misleading signals, and can you discuss the error rates and potential impacts in your training data?

- Why does pure code fine-tuning mostly outperform mixed training in RQ1?

---

> ### Author Response · Authors · 2025-11-22
> **Rebuttal - Reviewer DDwQ (1/2)**
>
> We thank Reviewer DDwQ for the constructive feedback and highlighting the comprehensiveness of our experiments. We hope to address your questions below.
>
> ### W1.& Q1. **Concern with pre-training effects mixing with intrinsic code properties.**
> We acknowledge the concern regarding pretraining effects. Since the exact pretraining data of these models is unknown, we are unable to disentangle intrinsic code properties from pretraining influences. Moreover, due to computational constraints, performing pretraining ourselves is infeasible. Therefore, analyzing pretraining effects versus code properties is beyond the scope of this work. To partially mitigate this limitation and reduce dependence on any single pretrained model, we evaluate models across multiple model families (Qwen3, LLaMA-3.2, Gemma3, OLMo2, SmolLM2) and scales.
>
> ### W2.1. & Q2. **Addressing Code Correctness Concerns**
>
> We appreciate the reviewer raising this concern and have substantially expanded our validation to address it. We now include both automated syntax/compilation checks across all ten programming languages (showing a mean pass rate of 82.59%) and a human evaluation of perturbation correctness, where 351/390 (90%) sampled instances were annotated as correct, including 84% accuracy for rule-based and 97% for generative perturbations. Collectively, these results demonstrate that the synthetic code is largely valid and that perturbations preserve their intended semantics. Moreover, because all perturbation variants share the same underlying GPT-4o-mini–generated base dataset, any remaining noise affects all conditions uniformly and cannot explain the consistent patterns observed across all experiments. Please see our deatailed response subsequenbtly.
>
> ### W2.2. **Concern with structural perturbations are overly simplified**
>
> We appreciate the reviewer raising this important concern and would like to clarify our validation methodology, which we have now expanded significantly in response to this feedback.
>
> #### **Validation of code correctness.**
> We have conducted extensive validation of our synthetic code data through two complementary approaches:
>
> **1. Syntax and compilation validation.**
> We performed automated syntax checking and compilation/execution testing across all ten programming languages in our 120K code instruction dataset. As shown in the table below, the vast majority of samples successfully compiled or executed, with pass rates ranging from **64.08% (TypeScript)** to **99.25% (Python)**, with a **mean pass rate of 82.59%** across all languages. This indicates that most instructions produce syntactically valid and executable code.
>
> | Language   | Total  | % Passed |
> |-----------|--------|----------|
> | Python    | 11993  | 99.25   |
> | PHP       | 12009  | 94.81   |
> | JavaScript| 11996  | 91.57   |
> | Java      | 12003  | 88.94   |
> | C++       | 11997  | 83.20   |
> | C         | 11998  | 81.49   |
> | C#        | 11996  | 81.06   |
> | Go        | 12012  | 77.77   |
> | Rust      | 11995  | 66.71   |
> | TypeScript| 12001  | 64.08   |
>
> **2. Manual perturbation quality evaluation.**
> We randomly sampled **30 examples per perturbation type** (13 types total: 7 rule-based and 6 generative) and manually verified correctness.
> - For **rule-based** perturbations and comment modifications, we ensured the transformation strictly followed the intended rule while leaving all other content unchanged.
> - For **generative** perturbations (pseudocode, step-by-step, flowchart, imaginary language), we verified that the conveyed semantics remained faithful to the original code.
>
> Across **390** sampled instances, **351** were correct (**90% overall**), with:
> - **84% correctness** for rule-based perturbations,
> - **97% correctness** for generative perturbations.
>
> ---

---

> ### Author Response · Authors · 2025-11-22
> **Rebuttal - Reviewer DDwQ (2/2)**
>
> #### **Addressing the test case concern.**
> Execution-based semantic validation is indeed the gold standard, but for 120K multi-language examples it is computationally infeasible. Instead, we conducted:
>
> 1. **Compilation/execution checks** provide strong syntactic validity.
> 2. **Manual perturbation checks** confirm semantic preservation.
> 3. Prior work has shown that **instruction-finetuning with code, even without execution tests, consistently improves reasoning** [1].
>
> Our study focuses on **which properties of code contribute to reasoning**, not on evaluating code generation correctness. This question remains meaningful even if not every single training example is perfect (passing all the desired test cases), as long as quality is consistent across perturbations, which our validation confirms.
>
> ---
>
> We believe our comprehensive validation approach, along with the systematic nature of our perturbation design and the consistency of results across **3,331 experiments**, provides strong evidence for the validity and reliability of our findings.
>
>
>
> ### W3. & Q3. **Discussion on why pure code fine-tuning mostly outperforms mixed fine-tuning.**
>
> We thank the reviewer for this insightful feedback and question. Our findings suggest that the superiority of pure code fine-tuning is more likely arisen from the nature of structured code signals rather than data imbalance or quality mismatches. Code provides deterministic, compositional, and strongly constrained supervision (e.g., loops, conditionals, dataflow), which closely aligns with the multi-step reasoning evaluated in math and code tasks. As shown in Figure 22 on Appendix page 23, models trained on pure code consistently outperform mixed variants on reasoning-heavy tasks. Since instruction data is curated from high-quality sources and filtered identically, and our synthetic code data and its perturbed variants additionally shows an 82.59% compilation pass rate and 90% human-verified perturbation correctness, data-quality differences alone cannot explain the observed trend.
>
> Our study focuses specifically on instruction fine-tuning, where all variants begin from the same pretrained checkpoint and share identical hyperparameters, only the data composition changes. Model performance comparisons under this setup isolate post-training effects and confounding pretraining factors. In this stage, mixed training introduces competing gradient signals from heterogeneous distributions.
> Pure code fine-tuning, by contrast, exposes the model to a single, structurally homogeneous training distribution, which reduces gradient interference and likely enables the model to converge toward a representation space that captures features beneficial for math and code reasoning, as shown in other work[1]. While mixed training may appear intuitively aligned with diverse evaluation tasks, our results show that the structural regularity of the training signal rather than variety of topics, plays a more dominant role in strengthening reasoning capabilities.
>
> [1] Zhang, Xinlu, et al. "Unveiling the impact of coding data instruction fine-tuning on large language models reasoning." AAAI 2025.
>
> ### Q1. **Concern with pre-training effects mixing with intrinsic code properties**
> Please provide our response for this in W1.
>
>
>
> ### Q2. **Regarding code correctness, error rates, and potential impacts**
>
> We provide a detailed response in W2.
>
>
> ### Q3. **Inadequate Analysis of Why Pure Code Fine-tuning Outperforms Mixed Training in RQ1**
>
> Please see our response for W3.

---

### Official Review · Reviewer_345W · 2025-11-01

**Soundness:** 3
**Presentation:** 4
**Contribution:** 3
**Rating:** 8
**Confidence:** 4

**Summary:**

The paper probes which properties of code drive LLM reasoning via 3,331 finetuning experiments across five model families, eight scales, ten languages, and systematic perturbations. It finds structural features are pivotal, with their disruption reliably hurting math and code performance. Abstractions and efficient encodings can rival raw code, and even corrupted code stays competitive when surface regularities persist. Lower level languages yield larger math gains, suggesting practical principles for training data design beyond executable programs.

**Strengths:**

1. The paper demonstrates, from multiple complementary perspectives, that code data confers genuine benefits to LLM reasoning, and the evidence supporting the central claim that structural properties are decisive is clear, consistent, and persuasive.
2. The experimental suite is broad in scope and methodologically solid, spanning diverse model families, sizes, languages, and perturbations, and it cleanly validates how targeted changes to structure or semantics translate into measurable differences in downstream performance.
3. Several results crystallize actionable insights for the community, pointing toward data design practices that emphasize structural signals and efficient abstractions, and suggesting concrete priorities for future training and evaluation.

**Weaknesses:**

1. It would be valuable to extend the study to larger model sizes to observe how the perturbations interact with scaling.
2. It is unclear whether the conclusions still hold in a reasoning model setting.

**Questions:**

Please refer to weakness.

---

> ### Author Response · Authors · 2025-11-22
> **Rebuttal - Reviewer 345W**
>
> We thank Reviewer 345W for the supportive assessment of our experimental framework and the impact of our findings. We hope to address your comments and suggestions below.
>
> ### W1. **Experiments on larger models**
>
> We appreciate this suggestion and acknowledge the importance of validating our findings at larger scales. Our experiments (span 0.6B to 8B) result in over 300 fine-tuning runs and thousands of H100/A100 GPU hours. Given that each experimental configuration requires 13 perturbation variants (plus 3 code/mixed/nl and 10 language-specific variants), scaling to 14B+ models presents significant computational challenges. Our estimates indicate that each 14B model requires ~60 H100 GPU hours, thus over 2,000 H100 GPU hours for a complete experimental sweep (accounting for reruns and evaluation), which exceeds our available compute budget.
> However, following your suggestion, we conducted targeted validation experiments with Qwen3-14B-Base to assess whether our key findings generalize to larger scales. Specifically, we trained the 14B model on:
>
> Baseline conditions: code-only (code-ft), natural language-only (nl-ft), and mixed (50-50) training (mixed) and 2 rule-based perturbations: swap comments local, replace keywords nonsense.
>
> Due to computational constraints, we used 30K training samples (vs. 120K in main experiments) and trained for 1 epoch with otherwise identical hyperparameters. We evaluated on CodeMMLU (code understanding), MMLU (general reasoning), and GSM8K (math reasoning):
>
>
>
>
> | Configuration               | CodeMMLU | MMLU   | GSM8K  |
> |---------------------------|----------|--------|--------|
> | zeroshot                  | 0.6567   | 0.7802 | 0.8658 |
> | code-ft                   | 0.6688   | 0.7697 | 0.8537 |
> | nl-ft                     | 0.6548   | 0.7787 | 0.8552 |
> | mixed                     | 0.6668   | 0.7710 | 0.8097 |
> | swap comments local       | 0.6453   | 0.7741 | 0.8567 |
> | replace keywords nonsense | 0.6372   | 0.7765 | 0.8127 |
>
>
> These results show that:
> (1) Code-only training remains strong on code and math tasks (CodeMMLU, GSM8K).
> (2) Perturbations still degrade performance, especially structural ones.
> (3) Task dependence persists—nl-ft slightly favors MMLU, while code-ft performs best on reasoning-heavy benchmarks.
>
>
> While these experiments on 14B models use less data (only 30K) and training duration (hence not directly comparable), the qualitative patterns align with our comprehensive findings, suggesting our core conclusions about code properties and reasoning generalize to larger models. We acknowledge that full-scale validation across all perturbations at 14B+ would strengthen our claims and note this as valuable future work.
>
> ### W2. **Hypothesized findings in a reasoning-model setting.**
> We thank the reviewer for this suggestion! This direction is beyond the scope of our current focus, but we will consider it for future work.

---

### Official Review · Reviewer_i4ZX · 2025-11-04

**Soundness:** 3
**Presentation:** 3
**Contribution:** 2
**Rating:** 4
**Confidence:** 4

**Summary:**

- The paper studies why code improves reasoning in LLMs through a large-scale, systematic framework.

- Constructs parallel instruction datasets in 10 programming languages and applies controlled perturbations (rule-based and generative) that selectively distort syntax, semantics, or structure.

- Finetunes 5 model families across 8 scales (3,331 experiments) and evaluates effects on natural language, math, and code reasoning tasks.

- Finds that structural features of code drive reasoning improvements more than semantics; pseudocode and flowcharts can substitute for code; lower-level languages aid math reasoning.

**Strengths:**

- Clever perturbation design isolating structure vs. semantics; highlights causal signals in code-induced reasoning.

- Impressive experimental scale and rigor - 3k+ controlled finetunes across models, languages, and perturbation types.

**Weaknesses:**

- Heavy reliance on GPT-4o-mini for generating and evaluating code; possible bias leakage or contamination

- Finetuning limited to ≤8B models; it's unclear if findings hold for frontier scales where reasoning mechanisms differ.

- Focus on finetuning setups - the model performance could depend on the pretrain LLMs.

- It would be great to have some ablations on dataset size or diversity. This would help disentangle whether effects persist with smaller or noisier samples.

**Questions:**

- How robust are results to using a different evaluator model (e.g., Claude, Gemini) for judging code quality?

- How would findings change if applied during pretraining rather than instruction finetuning?

- Could smaller-scale runs (e.g., 20k data) reproduce the same trends?

---

> ### Author Response · Authors · 2025-11-22
> **Rebuttal - Reviewer i4ZX (1/3)**
>
> We thank Reviewer i4ZX for their constructive feedback and for highlighting our novel perturbation design and experimental rigor. We appreciate the opportunity to address your comments below.
>
> ### W1. **Heavy reliance on GPT-4o-mini for generating and evaluating code**
>
> We appreciate this concern and have conducted comprehensive validation to address it. To test evaluation robustness, we assessed all Qwen3-4B model variants using five different LLM-as-judge models: claude-3-haiku-20240307, claude-haiku-4-5-20251001-v1_0, gpt-4o-mini, gpt-5-mini, and llama3-2-90b-instruct. While absolute scores varied across judges (6.58-9.33), model rankings remained highly consistent with moderate inter-judge agreement (std 0.63-0.99). Critically, the relative performance patterns and conclusions align across all evaluators, demonstrating that our findings are robust regardless of the chosen judge. The complete multi-judge results are provided in our detailed response.
>
> Regarding data generation, we emphasize that our conclusions focus on which properties of code affect reasoning, not on absolute code quality. All perturbations share the same GPT-4o-mini-generated base dataset, so any potential biases affect all variants uniformly. Our validation shows 82.59% mean compilation pass rate across languages and 90% correctness in manual evaluation of perturbations. The consistent patterns observed, e.g., structural perturbations degrading performance more than semantic ones, hold across five model families and eight scales (3,331 experiments), indicating these are properties of code training rather than GPT-4o-mini-specific artifacts.
>
>
> ### W2. **Finetuning limited to ≤8B models**
>
> We appreciate this suggestion and acknowledge the importance of validating our findings at larger scales. Our experiments (span 0.6B to 8B) result in over 300 fine-tuning runs and thousands of H100/A100 GPU hours. Given that each experimental configuration requires 13 perturbation variants (plus 3 code/mixed/nl and 10 language-specific variants), scaling to 14B+ models presents significant computational challenges. Our estimates indicate that each 14B model requires ~60 H100 GPU hours, thus over 2,000 H100 GPU hours for a complete experimental sweep (accounting for reruns and evaluation), which exceeds our available compute budget.
> However, following your suggestion, we conducted targeted validation experiments with Qwen3-14B-Base to assess whether our key findings generalize to larger scales. Specifically, we trained the 14B model on:
>
> Baseline conditions: code-only (code-ft), natural language-only (nl-ft), and mixed (50-50) training (mixed) and 2 rule-based perturbations: swap comments local, replace keywords nonsense.
>
> Due to computational constraints, we used 30K training samples (vs. 120K in main experiments) and trained for 1 epoch with otherwise identical hyperparameters. We evaluated on CodeMMLU (code understanding), MMLU (general reasoning), and GSM8K (math reasoning):
>
>
>
> | Configuration               | CodeMMLU | MMLU   | GSM8K  |
> |---------------------------|----------|--------|--------|
> | zeroshot                  | 0.6567   | 0.7802 | 0.8658 |
> | code-ft                   | 0.6688   | 0.7697 | 0.8537 |
> | nl-ft                     | 0.6548   | 0.7787 | 0.8552 |
> | mixed                     | 0.6668   | 0.7710 | 0.8097 |
> | swap comments local       | 0.6453   | 0.7741 | 0.8567 |
> | replace keywords nonsense | 0.6372   | 0.7765 | 0.8127 |
>
>
> These results show that:
> (1) Code-only training remains strong on code and math tasks (CodeMMLU, GSM8K).
> (2) Perturbations still degrade performance, especially structural ones.
> (3) Task dependence persists—nl-ft slightly favors MMLU, while code-ft performs best on reasoning-heavy benchmarks.
>
>
> While these experiments on 14B models use less data (only 30K) and training duration (hence not directly comparable), the qualitative patterns align with our comprehensive findings, suggesting our core conclusions about code properties and reasoning generalize to larger models. We acknowledge that full-scale validation across all perturbations at 14B+ would strengthen our claims and note this as valuable future work.

---

> ### Author Response · Authors · 2025-11-22
> **Rebuttal - Reviewer i4ZX (2/3)**
>
> ### W3. **Concern with the model performance depending on the pretrained LLMs**
>
> We agree that the foundations of the base pretrained models may influence downstream performance. To ensure fair comparison, our study was carefully designed in the following way:
> 1. **Controlled baseline across all perturbed variants.**
> Within each model family and scale, all experiments start from the same pretrained architecture. We first instruction-finetune this model on the full, unperturbed version of our synthesized 120K code instruction–response pairs. We then finetune the models on each perturbed variant. Therefore, performance comparisons across perturbations are grounded on an identical model initialization to ensure that the observed differences stem from perturbation effects rather than pretraining variability.
> 2. **Cross-family and cross-scale model evaluation.**
> To account for potential variability introduced by pretraining objectives or architecture differences, we evaluate models across multiple model families (Qwen3, LLaMA-3.2, Gemma3, OLMo2, SmolLM2) and scales. We hope this would address the concern of dependency on a specific pretrained model and improve the generality of our findings.
>
> Lastly, to computational constraints, we couldn't perform full pretraining from scratch. However, prior research has consistently demonstrated that incorporating code in the instruction-finetuning stage already leads to notable performance improvements [1]. This evidence supports the validity of conducting perturbation analysis at the IFT stage given the resource constraints.
>
>
> ### W.4. and Q.3. -- **Ablations on smaller dataset size.**
>
> Following reviewer's suggestion we train the Qwen3-4B-Base on 25% (30K) examples. All other parameters are kept identical to other training. We report the key findings in the table below and corresponding plots are in https://anonymous.4open.science/r/smaller_data_size_experiments-2DC7/. Notably, they align with our findings reported in the paper.
>
> | **Axis** | **Config** | **NL** | **Math** | **Code Understanding** | **Code Generation** |
> |---------|------------|--------|---------|------------------------|---------------------|
> | **structure vs semantics** | structure perturbations | 0.553 ± 0.0168 | 0.722 ± 0.0118 | 0.564 ± 0.00250 |  |
> |  | semantics perturbations | 0.564 ± 0.0169 | 0.774 ± 0.0112 | 0.587 ± 0.00250 |  |
> | **human interpretability** | high | 0.555 ± 0.0174 | 0.697 ± 0.0133 | 0.554 ± 0.00250 |  |
> |  | low | 0.560 ± 0.0165 | 0.779 ± 0.0104 | 0.589 ± 0.00250 |  |

---

> ### Author Response · Authors · 2025-11-22
> **Rebuttal - Reviewer i4ZX (3/3)**
>
> ### Q1. **Different LLM-as-judge for code generation evaluation.**
>
> To address this concern about the robustness of our evaluation methodology, we conducted a comprehensive analysis using five different state-of-the-art LLM-as-judge models: claude-3-haiku-20240307, claude-haiku-4-5-20251001-v1_0, gpt-4o-mini, gpt-5-mini, and llama3-2-90b-instruct. We evaluated all Qwen3-4B model variants on Python code generation tasks using identical evaluation settings across all judge models, with each judge assessing the same set of generated code samples. We present the mean scores and standard deviations across different target models for each judge in the table below.
>
>
>
> | **Target Model<br> + Perturbation** | claude-3-haiku-20240307 | claude-haiku-4-5-20251001-v1_0 | gpt-4o-mini | gpt-5-mini | llama3-2-90b-instruct | Mean of Judges | Std of Judges |
> |--------------|-------------------------|--------------------------------|-------------|------------|-----------------------|----------------|---------------|
> | zeroshot | 8.4098 ± 2.4215 | 7.3378 ± 2.9898 | 7.0854 ± 3.2699 | 6.8372 ± 3.0014 | 8.1707 ± 2.5921 | 7.57 | 0.69 |
> | swap_comments_global | 9.0065 ± 1.4072 | 7.7581 ± 2.4340 | 7.9085 ± 2.5497 | 6.8462 ± 2.9381 | 9.1299 ± 1.4175 | 8.13 | 0.95 |
> | swap_comments_local | 9.2446 ± 1.2444 | 7.9388 ± 2.6895 | 8.5366 ± 2.3664 | 7.2578 ± 3.0617 | 9.1504 ± 1.5787 | 8.43 | 0.84 |
> | replace keywords_nonsense | 9.0974 ± 1.1647 | 7.8817 ± 2.4577 | 8.7378 ± 2.3073 | 7.1648 ± 3.0414 | 9.2013 ± 1.3861 | 8.42 | 0.87 |
> | replace_keywords_nonEn | 9.2051 ± 1.1550 | 7.6767 ± 2.6983 | 8.7256 ± 2.2246 | 7.3500 ± 2.8865 | 9.2532 ± 1.2022 | 8.44 | 0.88 |
> | flowchart | 8.8780 ± 1.6718 | 7.4416 ± 2.8560 | 8.0732 ± 2.5825 | 7.3241 ± 3.0072 | 9.1275 ± 1.5035 | 8.17 | 0.82 |
> | imaginary | 8.9390 ± 1.6751 | 7.6735 ± 2.8674 | 8.0610 ± 2.7126 | 7.1714 ± 2.9354 | 9.0000 ± 1.8235 | 8.17 | 0.80 |
> | pseudocode | 8.8902 ± 1.6242 | 7.1468 ± 2.7177 | 7.3415 ± 2.9958 | 7.2199 ± 2.9010 | 9.0154 ± 1.4842 | 7.92 | 0.94 |
> | step_by_step | 8.6646 ± 1.9227 | 7.5000 ± 2.8273 | 7.7561 ± 2.7450 | 7.5692 ± 2.9040 | 8.8110 ± 1.9011 | 8.06 | 0.63 |
> | comment obfuscation | 8.7931 ± 2.0281 | 7.4091 ± 2.8033 | 7.8598 ± 2.8798 | 7.0217 ± 3.0705 | 8.9268 ± 1.7812 | 8.00 | 0.84 |
> | comment enhancement | 9.2195 ± 1.3523 | 7.7748 ± 2.7392 | 8.3598 ± 2.6466 | 7.9286 ± 2.4785 | 9.0854 ± 1.7103 | 8.47 | 0.66 |
> | remove_comments | 9.0854 ± 1.6106 | 7.8571 ± 2.5357 | 8.2683 ± 2.5383 | 7.2250 ± 3.0499 | 9.1879 ± 1.4207 | 8.32 | 0.83 |
> | remove_whitespace | 9.1006 ± 1.4284 | 7.8667 ± 2.5766 | 8.6890 ± 2.2824 | 7.7737 ± 2.8491 | 9.3284 ± 1.3252 | 8.55 | 0.71 |
> | replace_variables | 9.2000 ± 0.9506 | 7.4701 ± 2.7357 | 8.0915 ± 2.6604 | 6.9225 ± 3.0921 | 9.0677 ± 1.5334 | 8.15 | 0.99 |
> | code-ft | 9.1384 ± 1.4647 | 7.9796 ± 2.3228 | 8.6463 ± 2.3595 | 7.6091 ± 2.9803 | 9.1707 ± 1.4082 | 8.51 | 0.70 |
> | mixed-ft | 8.6768 ± 1.9375 | 7.4588 ± 2.9421 | 7.9390 ± 2.7884 | 7.3056 ± 3.0768 | 8.7500 ± 2.0973 | 8.03 | 0.67 |
> | nl-ft | 8.1869 ± 2.4307 | 6.5823 ± 2.9888 | 6.7500 ± 3.1082 | 6.5973 ± 3.1471 | 7.8409 ± 2.6322 | 7.19 | 0.76 |
>
>
> Our multi-judge evaluation demonstrates that model rankings remain highly consistent across all five judge models, despite variation in absolute scores (ranging from 6.58 to 9.33). Importanlty, the Mean of Judges rankings closely align with gpt-4o-mini rankings reported in our main paper. The Std of Judges shows moderate inter-judge agreement (std 0.63-0.99), which indicates that while absolute score calibration varies between evaluators, relative assessments remain stable. Together, these validate that our main paper's conclusions based on gpt-4o-mini are robust and would hold regardless of the chosen evaluator.
>
> ### Q2. **Hypothesized findings if applied during pre-training other than IFT**
> We appreciate the reviewer's suggestion. Extending perturbation analysis to the pretraining stage is an interesting direction, but is orthogonal to our setup and falls outside the scope of this work. We note that prior work such as [2] has investigated the role of code during pretraining, and we will reference this direction as a potential future study.
>
>
>
> ### Q3. **Could smaller-scale runs (e.g., 20k data) reproduce the same trends?**
>
> Following your suggestion, we train the models on 25% (30K examples) of whole data. We report the results and key findings in response to W4.
>
>
> [1] Zhang, Xinlu, et al. "Unveiling the impact of coding data instruction fine-tuning on large language models reasoning." Proceedings of the AAAI Conference on Artificial Intelligence. Vol. 39. No. 24. 2025.
> [2] Aryabumi, Viraat, et al. "To code, or not to code? exploring impact of code in pre-training." arXiv preprint arXiv:2408.10914 (2024).

---

> > ### Comment · Reviewer_i4ZX · 2025-11-27
> >
> > Thank you for adding new experiments and further clarification. I have increased my score from 4 to 5.
> >
> > I'd strongly suggest that the authors include some of the limitations discussed in the rebuttal (e.g., limited to relatively small post-trained models) in the updated paper, thereby strengthening the insights and takeaway.

---

### Official Review · Reviewer_zrYv · 2025-11-10

**Soundness:** 3
**Presentation:** 2
**Contribution:** 3
**Rating:** 6
**Confidence:** 4

**Summary:**

In this work, the authors set out to validate the claim that coding data enhances reasoning abilities of LLMs, while also deepening the analysis by **ablating how different aspects of coding data correlate with downstream evaluations across three categories of tasks**: Natural language and general knowledge, math, and code generation and understanding.

In particular, the authors collate prompts from various open-source code instruction-tuning benchmarks, and ask GPT-4o to generate completions in 10 popular programming languages, resulting in a final dataset of 120k samples (12k per each language). They then provide 11 perturbed versions of the original dataset - 5 variants the result of programmatic perturbations (e.g. removing whitespace, renaming variables, removing comments etc.), and 6 variants of LLM rewrites (e.g. comment enhancement, conversion to pseudocode, or to natural language).

Those datasets - along with a 120k general instruction following examples from OpenHermes 2.5 that constitute a non-code baseline - are then grouped into logical subcategories, and used to finetune models of various model families (e.g. Qwen3, Llama3, Gemma3), ranging from sizes of 0.6B to 8B.

The main findings:
- Incorporating (unperturbed) **code data generally improves performance** across all tasks, especially math and coding (Figure 1).
- **The perturbed datasets reduce model performance**, as compared to the unperturbed dataset.
- **Semantic perturbations reduce performance less than structural** (Figure 3).
- Perturbations which **highlight algorithmic structure** while removing superficial syntax, like pseudocode or flowcharts, often **match or surpass unperturbed code baseline** (Figure 4).
- Models remain **robust to corrupted or low-interpretability code** (Figure 6).

**Strengths:**

- Investigating the of usefulness of different aspects of code data is a **timely and relevant, with clear practical utility.**

- The work is **original in both framing and experimental design.** Disentangling different aspects of code data (structural vs semantic, human interpretable vs not) in the context of the data’s generalization to reasoning is a novel advance over prior, more-coarse grained (“code helps reasoning”) empirical investigations.

- The **experiments are extensive**, comprising 3,331 fine-tuning runs across ten programming languages, eleven perturbation types, five model families, and multiple model scales ranging from 0.6B to 8B parameters. The appendix presents these results exhaustively, providing detailed plots and breakdowns that reinforce the empirical trends highlighted in the main text.

- The paper **provides comprehensive replication details** in its appendix, including exact hyperparameters, dataset processing steps, and all prompt templates used for generation and evaluation.

**Weaknesses:**

- The **presentation of the perturbed datasets lacks detail** - the paper would be enhanced by a) providing overall statistics like the number of tokens, and b) providing qualitative examples of each dataset.

- All **experiments are restricted to small, up to 8B parameter models**, leaving doubt whether the observations would generalize to frontier-scale models. If possible, the authors could corroborate their main findings by a few experiments on larger models. Barring that, scaling laws could be established across the 1B -> 8B scale.

- Both the baseline and generative-perturbation code data are synthetically generated with GPT-4o (except for the prompts). However, **the paper provides no assessment of the quality and correctness of this generated data**. This could be mitigated by including a small-scale human evaluation—for example, manually annotating around 30 samples from each perturbation set. Alternatively, if resources permit, the authors could regenerate the data using a different frontier model and re-run key analyses to verify that the observed trends are consistent across data sources.

- The paper presents numerous ablations, but results are shown separately for each model–size combination, with only a single representative plot per research question in the main text. This makes it difficult to see aggregate patterns, as readers must cross-reference many individual plots in the appendix. **Aggregating results across models or scales would make the reported trends clearer and more convincing.**

- **Some claims in the main text are not fully substantiated by the experimental results.** For instance, when discussing performance across programming languages, the authors state that “on math tasks, high-scripting languages consistently underperform relative to intermediate and low-system ones,” referencing Figure 7 as evidence. However, in Figure 7 (top), high-scripting languages perform on par with intermediate ones and even surpass low-system languages on math tasks. In Figure 7 (bottom), the top five individual languages include three high-level ones, with all five well within each other’s error bars. Overall, the data shown (at least in the main body of the text) does not support the claim.

**Questions:**

See "Weaknesses" section

---

> ### Author Response · Authors · 2025-11-22
> **Rebuttal Reviewer zrYv (1/3)**
>
> We thank Reviewer zrYv for their careful reading and insightful suggestions, and pointing out the strength of our research question framing and experimental design. We hope to address each of the comments below.
>
> ### W1. **Details of the perturbed dataset**
> Thank you for the suggestion to provide details for each perturbation type. We have incorporated representative examples directly in Table 1 on page 4. Below, we additionally provide extended qualitative examples as well as the total token count and average token count per instruction for each perturbation variant in the dataset. We will include this table in an updated draft.
>
> | perturbation                        | original excerpt                         | perturbed excerpt                                | total token count | avg token count per instruction |
> |------------------------------------|-------------------------------------------|--------------------------------------------------|-------------------|----------------------------------|
> | whitespace removal                 | `for char in input_string:`               | `forcharininput_string:`                         | 78,553,430        | 654.61                           |
> | variable renaming                  | `for char in input_string:`               | `for var_4 in var_1:`                            | 87,619,500        | 730.16                           |
> | keyword replaced with non-sense    | `if c not in vowels:`                     | `garply c not in vowels:`                        | 87,123,587        | 726.03                           |
> | keyword replaced with non-English  | `if c not in vowels:`                     | `père c not in vowels:`                          | 88,078,906        | 733.99                           |
> | comment removal                    | `# Read input`                            | `-`                                              | 80,238,050        | 668.65                           |
> | local comment swapping             | `# Read input`                            | `# Process and print the result`                 | 85,324,436        | 711.04                           |
> | global comment swapping            | `# Process and print the result`          | `// Queue for processing nodes`                  | 85,329,862        | 711.08                           |
> | flowchart (Markdown)               | `if char not in vowels:`                  | `[Read char] → {Vowel?} → [Append '.' + lower]`  | 67,553,461        | 562.95                           |
> | step-by-step explanation           | `result.append('.' + char.lower())`       | `Append '.' before consonants …`                 | 84,250,378        | 702.09                           |
> | pseudocode                         | `for char in input_string:`               | `FOR EACH character IF not vowel THEN`           | 73,722,933        | 614.36                           |
> | imaginary language                 | `result.append('.' + char.lower())`       | `gloff add '.' ⊕ lower(chr)`                     | 81,011,032        | 675.09                           |
> | comment enhancement                | `# Process the result`                    | `# Removes vowels and prefixes consonants …`     | 119,399,621       | 994.99                           |
> | comment obfuscation                | `# Read input`                            | `# WARNING: Code may summon aliens ...`          | 111,771,640       | 931.43                           |

---

> ### Author Response · Authors · 2025-11-22
> **Rebuttal Reviewer zrYv (2/3)**
>
> ### W2. **Experiments on larger models.**
>
> We appreciate this suggestion and acknowledge the importance of validating our findings at larger scales. Our experiments (span 0.6B to 8B) result in over 300 fine-tuning runs and thousands of H100/A100 GPU hours. Given that each experimental configuration requires 13 perturbation variants (plus 3 code/mixed/nl and 10 language-specific variants), scaling to 14B+ models presents significant computational challenges. Our estimates indicate that each 14B model requires ~60 H100 GPU hours, thus over 2,000 H100 GPU hours for a complete experimental sweep (accounting for reruns and evaluation), which exceeds our available compute budget.
> However, following your suggestion, we conducted targeted validation experiments with Qwen3-14B-Base to assess whether our key findings generalize to larger scales. Specifically, we trained the 14B model on:
>
> Baseline conditions: code-only (code-ft), natural language-only (nl-ft), and mixed (50-50) training (mixed) and 2 rule-based perturbations: swap comments local, replace keywords nonsense.
>
> Due to computational constraints, we used 30K training samples (vs. 120K in main experiments) and trained for 1 epoch with otherwise identical hyperparameters. We evaluated on CodeMMLU (code understanding), MMLU (general reasoning), and GSM8K (math reasoning):
>
>
>
> | Configuration               | CodeMMLU | MMLU   | GSM8K  |
> |---------------------------|----------|--------|--------|
> | zeroshot                  | 0.6567   | 0.7802 | 0.8658 |
> | code-ft                   | 0.6688   | 0.7697 | 0.8537 |
> | nl-ft                     | 0.6548   | 0.7787 | 0.8552 |
> | mixed                     | 0.6668   | 0.7710 | 0.8097 |
> | swap comments local       | 0.6453   | 0.7741 | 0.8567 |
> | replace keywords nonsense | 0.6372   | 0.7765 | 0.8127 |
>
>
>
> These results show that:
> (1) Code-only training remains strong on code and math tasks (CodeMMLU, GSM8K).
> (2) Perturbations still degrade performance, especially structural ones.
> (3) Task dependence persists—nl-ft slightly favors MMLU, while code-ft performs best on reasoning-heavy benchmarks.
>
>
> While these experiments on 14B models use less data (only 30K) and training duration (hence not directly comparable), the qualitative patterns align with our comprehensive findings, suggesting our core conclusions about code properties and reasoning generalize to larger models. We acknowledge that full-scale validation across all perturbations at 14B+ would strengthen our claims and note this as valuable future work.
>
>
> ### W3.1. **Quality of synthetic code instructions.**
> As a first step, we conducted a syntax and compilation check across all ten programming languages. For each data instance, we extracted the code block from the instruction–response pair, wrote it to a temporary file with the correct language-specific extension, and applied language-appropriate syntax/compile checks (e.g., ast.parse for Python, gcc -fsyntax-only for C, javac for Java). As shown in the table below, the majority of samples successfully compiled or executed, indicating that most instructions are syntactically valid and runnable code.
>
> | Language     | Total | % Passed |
> |--------------|-------|----------|
> | C            | 11998 | 81.49%   |
> | PHP          | 12009 | 94.81%   |
> | JavaScript   | 11996 | 91.57%   |
> | Python       | 11993 | 99.25%   |
> | C++          | 11997 | 83.20%   |
> | TypeScript   | 12001 | 64.08%   |
> | Rust         | 11995 | 66.71%   |
> | C#           | 11996 | 81.06%   |
> | Go           | 12012 | 77.77%   |
> | Java         | 12003 | 88.94%   |
>
>
>
> ### W3.2. **Human Evaluation of Correctness of Perturbation sets.**
> As suggested by the reviewer we manually evaluated (2 human evaluators) the perturbation quality by randomly sampling 30 examples per perturbation (13 perturbation types in total: 7 rule-based and 6 generative).
>
> For the rule-based perturbations and comment enhancement/obfuscation, we verify that the transformation strictly follows the corresponding perturbation rule while keeping all other content unchanged.
> For other generative perturbations (pseudocode, step-by-step instructions, flowchart, imaginary language), which represent the original code in alternative forms, we verify that the conveyed semantics remain faithful to the original code.
>
> Across all 390 sampled instances, 351 were annotated as correct (90% overall). Specifically, the rule-based perturbations achieved 176/210 ≈ 84% correctness, while the generative perturbations achieved 175/180 ≈ 97% correctness.

---

> ### Author Response · Authors · 2025-11-22
> **Rebuttal Reviewer zrYv (3/3)**
>
> ### W4. **Aggregating results across models or scales.**
> We thank the reviewer for the suggestion. To this end, we further aggregated the results within each model family as well as all models across families and scales. We observed that the trends apply in the aggregated setting too. For example, structural perturbations are overall more detrimental than semantic ones, especially on math and code tasks; models trained on token-reduced perturbation data can perform well as long as core information is preserved; and LLMs can reason effectively from corrupted code that is low human-interpretable by exploiting surface-level regularities. We showcase the example results for Qwen3 family and all models below. The corresponding figures are in https://anonymous.4open.science/r/example_figures-82F0/.
>
> 1. Example results when aggregating within Qwen3 family.
>
> | **Axis** | **Config** | **NL** | **Math** | **Code Understanding** | **Code Generation** |
> |---------|------------|--------|---------|------------------------|---------------------|
> | **structure vs semantics** | structure perturbations | 0.510 ± 0.018 | 0.565 ± 0.014 | 0.540 ± 0.002 | 7.205 ± 0.094 |
> |  | semantics perturbations | 0.504 ± 0.019 | 0.596 ± 0.014 | 0.549 ± 0.002 | 7.486 ± 0.091 |
> | **human interpretability** | high | 0.502 ± 0.018 | 0.542 ± 0.014 | 0.527 ± 0.002 | 7.135 ± 0.094 |
> |  | medium | 0.508 ± 0.019 | 0.659 ± 0.015 | 0.564 ± 0.002 | 7.489 ± 0.092 |
> |  | low | 0.508 ± 0.018 | 0.576 ± 0.014 | 0.550 ± 0.002 | 7.434 ± 0.091 |
> | **relative information density** | strongly reduced  | 0.509 ± 0.018 | 0.519 ± 0.014 | 0.556 ± 0.003 | 7.039 ± 0.095 |
> |  | moderately reduced | 0.511 ± 0.019 | 0.620 ± 0.014 | 0.538 ± 0.002 | 7.355 ± 0.093 |
> |  | near-baseline | 0.498 ± 0.018 | 0.534 ± 0.013 | 0.547 ± 0.002 | 7.260 ± 0.091 |
> |  | increased | 0.505 ± 0.019 | 0.607 ± 0.014 | 0.547 ± 0.002 | 7.619 ± 0.091 |
>
> 2. Example results when aggregating all models.
>
> | **Axis** | **Config** | **NL** | **Math** | **Code Understanding** | **Code Generation** |
> |---------|------------|--------|---------|------------------------|---------------------|
> | **structure vs semantics** | structure perturbations | 0.444 ± 0.018 | 0.304 ± 0.012 | 0.370 ± 0.002 | 5.100 ± 0.083 |
> |  | semantics perturbations | 0.444 ± 0.018 | 0.325 ± 0.012 | 0.400 ± 0.002 | 5.216 ± 0.085 |
> | **human interpretability** | high | 0.442 ± 0.018 | 0.305 ± 0.012 | 0.371 ± 0.002 | 4.814 ± 0.081 |
> |  | medium | 0.447 ± 0.018 | 0.346 ± 0.012 | 0.405 ± 0.002 | 5.530 ± 0.088 |
> |  | low | 0.443 ± 0.018 | 0.311 ± 0.012 | 0.393 ± 0.002 | 5.194 ± 0.084 |
>
>
>
> ### W5. **Inconsistency between the main text and representative plots.**
> We appreciate the reviewer drawing attention to this. The figures (each shown for a single model) in the main text were intended as representative examples, while the analysis highlights general trends observed across all model families and scales. For Figure 7 on math task, the differences between high scripting and other variants are indeed small. We agree that the narrative could be more precise, and we will revise the wording to "most of the models on math tasks" in an updated draft to better align with the quantitative results. Alternatively, we could aggregate the results and replace Figure 7 with a consolidated trend.

---

### Author Response · Authors · 2025-12-04

## Summary
---
We sincerely thank all reviewers for their thoughtful feedback and constructive suggestions. Reviewers recognized the importance of our research questions, the rigor of our experimental design, and the practical insights our findings provide. We have incorporated reviewers' suggestions by adding extensive dataset details, validating data quality through compilation statistics and human evaluation, conducting additional experiments on larger models (Qwen3-14B), performing smaller-scale ablations, experimenting with multiple LLM-as-Judge models, and expanding aggregated trend analyses. We have revised the manuscript with all newly added content highlighted in blue.

Prior to the discussion period, reviewers provided scores of 6 (zrYv), 4 (i4ZX), 8 (345W), and 4 (DDwQ). **Reviewer i4ZX responded to our clarifications and raised their score from 4 to 6**, while the other reviewers did not have the opportunity to look into our rebuttal. Thus, **the updated scores are 4, 6, 6, and 8**. The majority of reviewers provided positive assessments, highlighting the comprehensiveness of our experimental setting and usefulness of our findings. Below, we summarize the reviews and our responses.

---
### For Reviewer zrYv (Score: 6)

Reviewer zrYv highlighted the timeliness and practical utility of our research question and the comprehensiveness of our experimental design. Their concerns focused on dataset details, experiments on larger models, data quality validation, result aggregation, and alignment between claims and figures.

**Response:** We added extended qualitative examples, token statistics, and detailed descriptions for all 13 perturbation types. We conducted compilation checks across ten languages (82.59% average pass rate) and human evaluation of perturbation correctness (90% overall accuracy). We performed targeted Qwen3-14B-Base experiments showing consistent patterns. We aggregated results within model families and across all models, confirming trends hold at aggregate levels. We clarified narrative precision regarding Figure 7.

---
### For Reviewer i4ZX (Score: 6, increased from 4)

Reviewer i4ZX praised our perturbation design and experimental rigor while raising concerns about reliance on GPT-4o-mini, limitation to ≤8B models, dependence on pretrained LLMs, and need for dataset size ablations.

**Response:** We conducted multi-judge evaluation using five state-of-the-art LLMs, demonstrating model rankings remain highly consistent across evaluators (std 0.63-0.99). We performed targeted 14B experiments showing consistent qualitative patterns. We clarified our controlled design where all perturbations share the same base model and dataset. We conducted 25% dataset ablations confirming all key findings hold. **The reviewer acknowledged these additions and increased their score from 4 to 6**.

---
### For Reviewer 345W (Score: 8)

Reviewer 345W provided highly supportive assessment, noting our work "crystallizes actionable insights for the community" with "clear, consistent, and persuasive" evidence. Their suggestions were extending to larger models and exploring reasoning-model settings.

**Response:** We conducted targeted Qwen3-14B-Base validation experiments showing our findings generalize to larger scales. We acknowledged the reasoning-model direction as valuable future work beyond our current scope.

---
### For Reviewer DDwQ (Score: 4)

Reviewer DDwQ acknowledged comprehensive experiments but raised concerns about pretraining effects versus code properties, code correctness and oversimplified perturbations, and why pure code outperforms mixed training.

**Response:** We clarified that disentangling pretraining effects requires retraining from scratch (computationally infeasible), but we mitigated this by evaluating across five model families. We provided extensive validation: 82.59% average compilation pass rate and 90% human-verified perturbation correctness (84% rule-based, 97% generative). We explained our perturbations were deliberately designed to isolate specific properties while maintaining cross-language generality, with all variants sharing the same base dataset. We provided detailed analysis showing pure code's superiority stems from structural regularity and reduced gradient interference, supported by prior work and our results.

---

**In summary**, we have substantially strengthened our paper through extensive additional experiments and validation, directly addressing every major concern.

---

### Meta-Review · Area_Chair_yDen · 2026-01-07

**Summary:**

Reviewers commended the paper's systematic perturbations isolating code properties' impact on LLM reasoning, its massive experimental scale, and actionable insights (e.g., structure > semantics, abstractions rival code).

Concerns included limited scale (≤8B), unvalidated code quality (potential errors/hallucinations), oversimplified perturbations, and unclear causal claims (pretraining confounds).

Rebuttals added validations (compilation/human checks, multi-judge, 14B runs, aggregations), and I think they are sufficient to address the concerns. Therefore, I think this paper should be accepted.

**Reviewer Concerns:**

The authors have addressed several key concerns: they provided dataset details including examples and token statistics, validated their approach on larger 14B models, demonstrated strong code quality (82.59% compilation rate, 90% human correctness), presented aggregate trends across model families, explained why code outperforms mixed data through gradient interference and regularity arguments, and showed robust rankings across multiple judges.

However, some questions remain unresolved: the effects of pre-training versus inherent code properties are unclear without from-scratch training runs, perturbation analyses lack depth (missing refactoring and execution tests), smaller-scale ablations were only partial (30K samples), and hypotheses about pure code's superiority.

**Reviewer Scores:**

- Reviewer zrYv: Would raise to 7 (rebuttals add details/validations but larger-scale needed).
- Reviewer i4ZX: Would maintain 6 (new experiments solid but pretraining limits remain).
- Reviewer 345W: Would maintain 8 (14B extension helpful).
- Reviewer DDwQ: Would maintain 4 (validations mitigate but causal claims flawed).

---

### Decision · Program_Chairs · 2026-01-26

Accept (Poster)